# GLEN-Bench: A Graph-Language based Benchmark for Nutritional Health

## Abstract

Nutritional interventions are important for managing chronic health conditions, but current computational methods provide limited support for personalized dietary guidance. We identify three key gaps: (1) dietary pattern studies often ignore real-world constraints such as socioeconomic status, comorbidities, and limited food access; (2) recommendation systems rarely explain why a particular food helps a given patient; and (3) no unified benchmark evaluates methods across the connected tasks needed for nutritional interventions. We introduce GLEN-Bench, the **first comprehensive g**raph-**l**anguage based b**e**nchmark for **n**utritional health assessment. We combine NHANES health records, FNDDS food composition data, and USDA food-access metrics to build a knowledge graph that links demographics, health conditions, dietary behaviors, poverty-related constraints, and nutrient needs. We test the benchmark using opioid use disorder, where models must detect subtle nutritional differences across disease stages. GLEN-Bench includes three linked tasks: risk detection identifies at-risk individuals from dietary and socioeconomic patterns; recommendation suggests personalized foods that meet clinical needs within resource constraints; and question answering provides graph-grounded, natural-language explanations to facilitate comprehension. We evaluate these graph-language approaches, including graph neural networks, large language models, and hybrid architectures, to establish solid baselines and identify practical design choices. Our analysis identifies clear dietary patterns linked to health risks, providing insights that can guide practical interventions. We release the benchmark with standardized protocols and evaluation tools at `https://anonymous.4open.science/r/GLEN-Benchmark-8B21`.

## 1 Introduction

Assessing nutritional health matters for both precision medicine and population health because diet is a modifiable factor that affects disease risk, symptom burden, and recovery (Armand et al., 2024; Berisha et al., 2025). For common conditions such as diabetes, cardiovascular disease, obesity, and kidney disease, diet works together with comorbidities, medications, and social determinants of health (SDoH) to influence patient outcomes (Maroto-Rodriguez et al., 2025; Simancas-Racines et al., 2025). While nutrition-aware assessment is universally acknowledged, existing computational efforts remain fragmented and rarely integrated into end-to-end decision workflows. Population-scale dietary and clinical resources, complemented by SDoH signals (e.g., access and affordability), offer the ingredients for nutrition-aware modeling (Seligman et al., 2023). Using these signals, recent benchmarks and datasets have focused on specific tasks, including food recommendation (Zhang et al., 2025b), diet logging and meal planning (Li et al., 2024c), risk prediction based on structured records (Zhang et al., 2024b), and nutrition-related question answering (Zhang et al., 2024a; Coburn et al., 2025), which typically support comprehensive assessment in a single environment.

However, turning these advances into practical, nutrition-aware interventions requires an integrated view that current resources still lack. In practical healthcare settings, effectively implementing dietary interventions involves: (i) identifying individuals who are likely to benefit from dietary interventions; (ii) recommending foods that are clinically appropriate and feasible within individual constraints (e.g., affordability and accessibility); and (iii) offering lucid explanations to build trust and improve adherence. Yet these resources

remain fragmented across modalities and task scopes, limiting end-to-end evaluation under realistic clinical and socioeconomic constraints.

Despite progress on individual tasks, current benchmarks do not support end-to-end evaluation of nutrition-aware interventions under realistic clinical and socioeconomic constraints. First, nutritional decision-making inherently involves multidimensional reasoning over dietary behaviors, health conditions, socioeconomic factors, and food access barriers; yet prior work often models these dimensions in isolation (e.g., nutrient targets without feasibility constraints). Second, tasks such as food recommendation and health risk detection are usually studied in isolation, even though they depend on one another in real-world settings. Third, the field lacks a standard evaluation "yardstick". Fragmented data sources and inconsistent metrics prevent direct comparisons across methods, making it difficult to identify which approaches work in real-world interventions.

Figure 1: An Overview of GLEN-Bench. Our framework brings scattered nutrition and health signals together into one pipeline for nutritional health assessment.

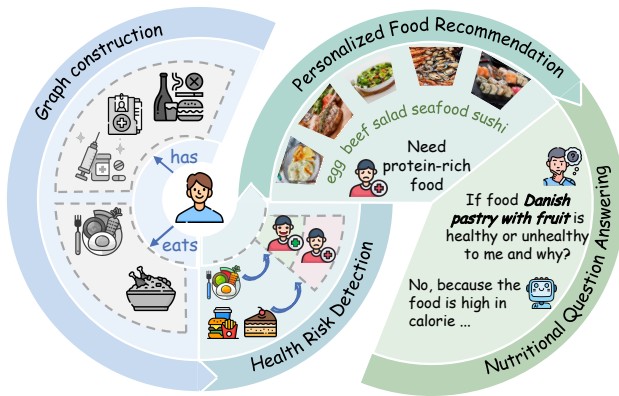

To bridge these gaps, we introduce GLEN-Bench, the first comprehensive **g**raph-**l**anguage based b**en**chmark for **n**utritional health assessment, supporting both structured graph reasoning and natural-language decision support. We build a disease-agnostic heterogeneous graph by integrating NHANES (CDC, 2024), FNDDS (Montville et al., 2013), and USDA access signals (United States Department of Agriculture, 2024). This structure captures essential dependencies between users, foods, and socioeconomic barriers. The schema is highly scalable; it extends to conditions like diabetes or obesity by simply adding condition-specific requirements. Furthermore, GLEN-Bench centers on an interdependent task suite that reflects end-to-end clinical workflows. Finally, we benchmark over 20 computational approaches, spanning classical ML, graph neural networks, large language models, and hybrid architectures, to establish a systematic baseline under a unified protocol.

We set up three tasks in our benchmark testing: (1) **Risk detection** identifies at-risk populations through dietary, social, and economic patterns. This not only supports population screening but also reveals how the model learns signals related to nutrition and health. (2) **Personalized recommendation** suggests foods that meet clinical guidelines and individual needs while remaining affordable and accessible. (3) **Question answering** assesses food-user compatibility via nutrient-level attribution and natural language explanations grounded in the knowledge graph, supporting transparent, informed decisions. These interdependent tasks form real clinical workflows, i.e., detection identifies intervention candidates, recommendation formulates tailored suggestions, and explanation ensures uptake. Beyond these core tasks, our design is scalable and naturally supports meal planning, ingredient substitution, dietary adherence prediction, and food-drug interaction analysis, making GLEN-Bench an evolving AI platform for nutritional intelligence. In summary, the main contributions are as follows:

- **First Comprehensive Multi-Task Benchmark for Nutritional Health**. We introduce the first comprehensive multi-task benchmark for nutritional health, which jointly evaluates risk detection, personalized recommendations, and explainable interpretations. GLEN-Bench is designed as a long-term scalable platform applicable to various nutrition-related diseases and supporting a wide range of downstream tasks.

- **A Novel Multi-Dimensional Nutrition Health Graph**. We built a large-scale nutrition-health graph to capture the complex inner relationships between diet, health, and social factors. This graph includes not only food and disease data, but also socioeconomic factors such as food insecurity, poverty, and access to healthcare. While previous datasets mostly focused on the association between nutrients and diseases, our approach provides a more comprehensive perspective.

- **Systematic Evaluation Revealing Design Principles**. We conducted the first systematic evaluation of over 20 methods, encompassing classic machine learning (ML), graph neural networks (GNNs), large language models (LLMs), and hybrid architectures. Our findings highlight the advantages of graph-based reasoning methods, where approaches combining graph neural networks with large language models achieved the best overall performance and more consistent interpretations under standardized protocols. We also further revealed interpretable dietary and socioeconomic characteristics associated with health risks, providing practical insights for developing effective interventions.

## 2 Related Work

**Computational Nutritional Health.** Computational methods for health risk assessment typically use electronic health records (EHRs), wearable devices, or survey data to predict chronic diseases and adverse health outcomes (Li et al., 2024a; Wu et al., 2010; Kasartzian & Tsiampalis, 2025). Some studies also analyze online behavior to identify at-risk populations and lifestyle patterns (Liu & Wang, 2025; Paul, 2025). However, most of these methods focus on prediction accuracy rather than providing clear explanations that can guide practical interventions. Research in public health has consistently shown connections between dietary patterns and disease risk, recovery outcomes, and socioeconomic inequalities (Berisha et al., 2025; Simancas-Racines et al., 2025; Maroto-Rodriguez et al., 2025; Wang et al., 2024a). These findings have driven the development of computational tools that integrate nutrition and behavioral data (Zhang et al., 2024b; Armand et al., 2024; Salinari et al., 2023). Food recommendation has progressed from preference-based ranking toward health-aware personalization (Li et al., 2023b; Shi et al., 2023; Tian et al., 2022a;b), evolving from single-metric optimization (e.g., calories or fat) (Ge et al., 2015; Shi et al., 2023) to integrating nutritional standards and health signals (Wang et al., 2021a; Bölz et al., 2023; Zhang et al., 2024b; 2025b; Shi et al., 2025). Nevertheless, many systems still overlook structural feasibility constraints (e.g., affordability and limited access) that shape real dietary choices. Knowledge graphs have been used to model food–health relationships for dietary adaptation and ingredient substitution (Haussmann et al., 2019; Chen et al., 2021; Fatemi et al., 2023; Forouzandeh et al., 2024), and recent KGQA and graph-RAG advances combine LLMs with GNNs to support structured reasoning (Jiang et al., 2023; Kim et al., 2023; Gao et al., 2024; Huang et al., 2026; Wen et al., 2024; Ju et al., 2022; Zhang et al., 2025a). Despite these advances, personalized services are often limited by a lack of user-specific medical information and socioeconomic factors (Bölz et al., 2023), and evaluation results remain inconsistent due to the lack of uniform, intervention-aligned benchmarks that reflect the end-to-end nutritional health workflow.

**Nutritional Health Benchmarks and Datasets.** Existing nutrition and health resources cover a variety of different modalities and task scenarios, including food-centric corpora, nutritional knowledge bases/ontologies, and health-oriented question-answering datasets. Recipe1M+ (Marın et al., 2021) provides large-scale multimodal recipes for food understanding and recommendation, while Nutrition5k (Thames et al., 2021) supports vision-based nutritional analysis and provides fine-grained nutritional component annotation. Structured food knowledge is represented by nutrition knowledge graphs (KGs) such as FoodKG (Haussmann et al., 2019) and standardized ontologies such as FoodOn (Dooley et al., 2018). In terms of nutrition-related question answering, RecipeQA (Yagcioglu et al., 2018) focuses on multimodal procedural reasoning, while other resources integrate nutritional ontologies and health indicators (Li et al., 2023a; Seneviratne et al., 2021) to enable structured food-health queries. Meanwhile, general graph reasoning and retrieval-augmented generation benchmarks advance multi-hop querying and structured reasoning (Li & Ji, 2022; Lewis et al., 2020; Besta et al., 2024), but remain largely generic or disconnected from clinical intervention contexts (Li et al., 2024b; He et al., 2024). Recent domain-specific datasets begin connecting diet and health contexts (Zhang et al., 2024b; 2025b; 2024a), yet they typically evaluate isolated tasks. Unlike prior benchmarks, we evaluate an end-to-end workflow with explicit feasibility constraints (e.g., affordability and access). For completeness, we include additional related work in the Appendix E.

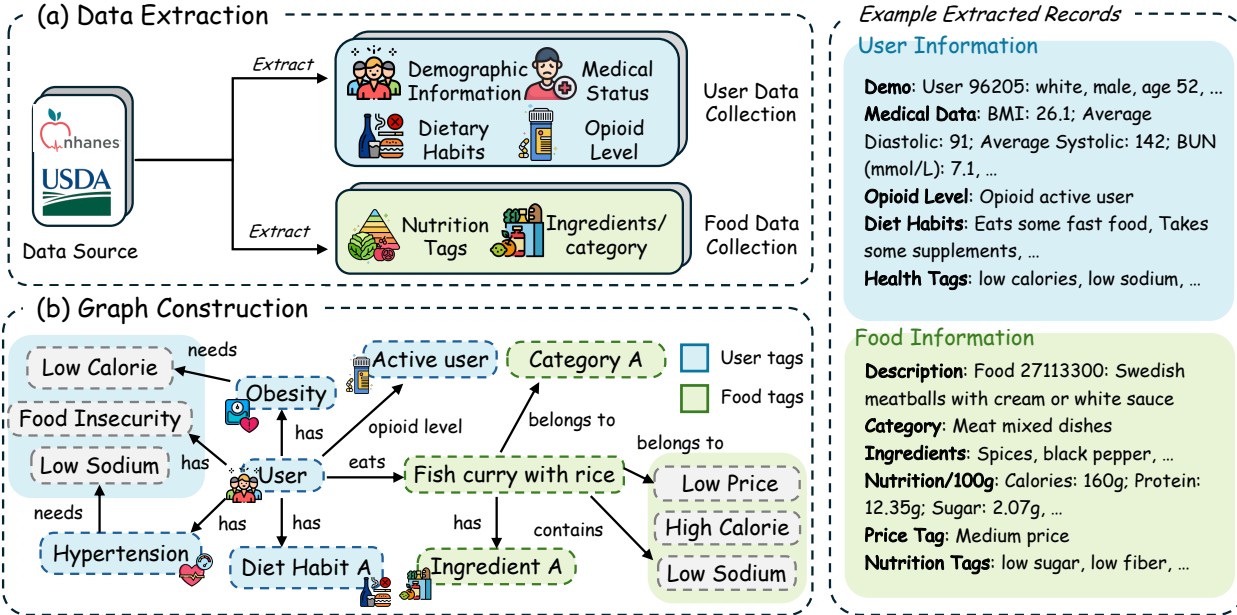

Figure 2: **The GLEN-Bench construction process.** (a) Data extraction from NHANES and USDA sources to collect multi-dimensional user and food information. (b) Graph construction of a heterogeneous knowledge graph modeling the relationships between clinical health status, dietary habits, socioeconomic barriers (e.g., food insecurity), and food nutritional profiles.

## 3 GLEN-Bench

### 3.1 Data Sources

GLEN-Bench integrates three population-level datasets to capture nutrition, health, and socioeconomic factors. From NHANES (CDC, 2024), we extract demographics, clinical data, medical conditions, medications, dietary behaviors, and 24-hour food intake records, along with socioeconomic indicators like income and food security status. Using FNDDS/WWEIA (Montville et al., 2013), we map foods to nutrient profiles and standardized categories, enabling nutrient-based food representations and threshold-defined nutrition tags. Finally, we augment foods with affordability tiers using USDA Purchase-to-Plate (United States Department of Agriculture, 2024) price estimates, allowing models to reason about economic feasibility alongside clinical suitability. The detailed source descriptions are provided in Appendix A.1.

### 3.2 Graph Construction

We utilize NHANES data from 2003 to 2020 to construct the GLEN Nutritional-Health Graph, a heterogeneous graph that captures the interactions between users, foods, and their clinical, behavioral, and socioeconomic backgrounds.

*Definition 3.1 GLEN Nutrition-Health Graph.* We model the graph as a directed heterogeneous graph $\mathcal{G} = (\mathcal{V}, \mathcal{E}, \mathcal{T}, \mathcal{R})$, where $\mathcal{V}$ is the set of nodes, $\mathcal{E} \subseteq \mathcal{V} \times \mathcal{V}$ is the set of edges, and $\mathcal{T}$ and $\mathcal{R}$ denote node and relation types. Each node $v \in \mathcal{V}$ and edge $e \in \mathcal{E}$ is assigned a type through mappings $\tau : \mathcal{V} \to \mathcal{T}$ and $\varphi : \mathcal{E} \to \mathcal{R}$. Node types include *user*, *food*, *ingredient*, *category*, *dietary_habit*, *health_condition*, *nutrition_tag*, *financial_tag*, and *price_tag*, while edges are used to describe consumption, ingredient composition, health and dietary habit context, and nutrition/price constraints.

**User Nodes.** Let $\mathcal{U} = \{v \in \mathcal{V} \mid \tau(v) = \mathsf{user}\}$ and $\mathcal{F} = \{v \in \mathcal{V} \mid \tau(v) = \mathsf{food}\}$. Each user node $u \in \mathcal{U}$ is associated with a feature vector $\mathbf{x}_u \in \mathbb{R}^{d_u}$ that concatenates demographic attributes (e.g., age, sex, race) with clinical and laboratory measurements (e.g., BMI, blood pressure, metabolic biomarkers) from NHANES.

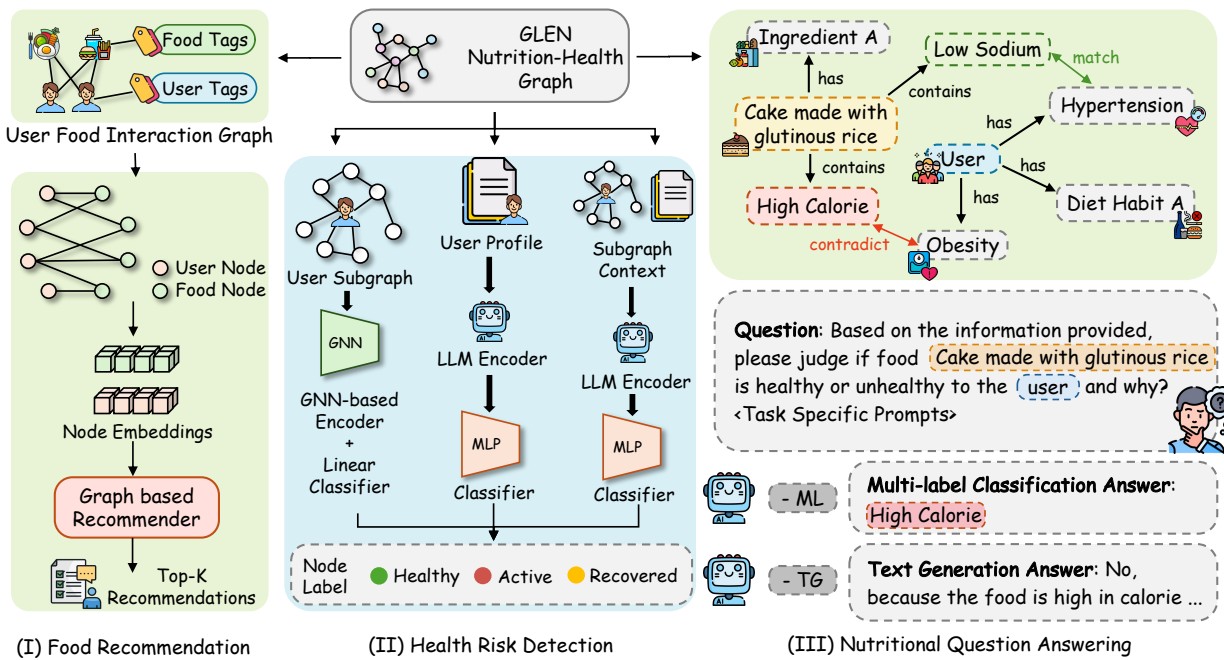

Figure 3: Overview of the GLEN-Bench framework. The system builds on a unified nutrition-health Graph to perform three related tasks: (I) personalized food recommendation, (II) health risk detection to identify intervention needs, and (III) nutritional question answering that provides explainable, graph-based explanations.

Diagnosed conditions, self-reported dietary habits, and financial hardship are modeled as explicit neighbors rather than entries in $\mathbf{x}_u$: users connect to *health_condition*, *dietary_habit*, and *financial_tag* nodes (e.g., poverty status and food insecurity). Together, these attributes and links collectively describe the medical and social environment that influences dietary behavior and vulnerability.

**Food Nodes.** Each food node $f \in \mathcal{F}$ has a feature vector $\mathbf{x}_f \in \mathbb{R}^{d_f}$ derived from FNDDS and USDA Purchase-to-Plate. From FNDDS and WWEIA we obtain nutrient profiles summarizing macro- and micronutrients, while *ingredient* and *category* nodes capture composition and hierarchical food groups; foods connect to these nodes via *contain* and *belong_to* edges. From Purchase-to-Plate we obtain price per 100 grams and discretize it into *price_tag* nodes (e.g., *low_price*/*medium_price*/*high_price*). Thus, each food is grounded by nutrient content and affordability, with additional context provided by ingredient and category neighbors.

**Health Conditions and Nutrition Tags.** Health-related dietary requirements are encoded by *health_condition* and *nutrition_tag* nodes. Health conditions are derived from NHANES clinical and questionnaire records and linked to users through edges. Nutrition tags represent guideline-based threshold constraints (e.g., *low_sodium*, *high_protein*); foods satisfying the thresholds connect via *food_has_tag* edges. We further associate conditions with relevant tags (e.g., hypertension with low sodium) to support downstream suitability assessment. Financial hardship is represented by *financial_tag* nodes (poverty and food insecurity) and considered jointly with *price_tag* nodes when modeling economic access.

**Edges and Relations.** Edges in $\mathcal{E}$ capture observed behavior and constraint structure: users link to consumed foods, conditions, habits, and financial tags, while foods link to ingredients, categories, nutrition tags, and price tags. Together, these relations encode what people eat, how foods are composed and priced, and how clinical status and socioeconomic constraints shape nutritionally appropriate and affordable options. The resulting graph integrates dietary behavior, health status, financial hardship, and food composition in a single heterogeneous structure. Let $\mathbf{X}^{\mathcal{U}} \in \mathbb{R}^{|\mathcal{U}| \times d_u}$ and $\mathbf{X}^{\mathcal{F}} \in \mathbb{R}^{|\mathcal{F}| \times d_f}$ denote the feature matrices for user and food nodes, and let $\mathbf{A}$ denote the adjacency structure induced by $\mathcal{E}$. Together, $(\mathcal{G}, \mathbf{X}^{\mathcal{U}}, \mathbf{X}^{\mathcal{F}})$ provide the

backbone for GLEN-Bench detection, recommendation, and question answering tasks across diverse health conditions. Details on feature processing and dataset statistics are provided in Appendix A.

**Tagging Scheme.** GLEN encodes clinical dietary requirements through two mechanisms. The first assigns nutrition tags to foods based on nutrient thresholds. For example, a food receives the `low_sodium` tag if its sodium content is at most 120 mg per 100g, and the `high_protein` tag if its protein content reaches 15g or above. This process yields 52,582 Food–NutritionTag edges across 18 tag types. The second mechanism connects health condition nodes to recommended nutrition tags, capturing domain knowledge such as hypertension requiring low sodium intake, or diabetes requiring low sugar and high fiber. Table 6 lists all condition-to-tag mappings used in GLEN. These two mechanisms work together to create interpretable constraint paths that recommendation and question answering tasks can use to produce clinically grounded outputs.

## 4 Nutritional Health Applications

This section instantiates GLEN-Bench's three core evaluation tasks and reports a standardized benchmarking study of diverse model families. For each task, we describe the task definition, evaluation metrics, baseline models and setup, and present the corresponding results and analyses.

### 4.1 Health Risk Detection

We apply GLEN-Bench to opioid use disorder (OUD) as an instance of health risk detection, requiring models to discern subtle nutritional and behavioral differences across active use, recovery, and health. The opioid crisis has claimed over 500,000 U.S. lives since 1999 (Hoffman et al., 2019; Herring et al., 2024), yet nutrition-informed support remains understudied despite evidence of poor diet quality and socioeconomic hardship among individuals with OUD (Cunningham, 2016; Waddington et al., 2023; Heflin & Sun, 2022; Chavez & Rigg, 2020; Nagarajan & Goodman, 2020; Wiss, 2019). Accordingly, we include OUD risk detection to evaluate screening under realistic clinical and socioeconomic constraints.

**Task Definition.** The health risk detection task aims to classify each user node in the GLEN Nutrition-Health Graph into health-status categories based on dietary and socioeconomic signatures. For our OUD instantiation, this is formulated as a three-class classification problem: *active opioid user*, *opioid-recovered user*, or *normal user*. Unlike binary settings, we formulate a three-class problem and leverage multi-relational graph context. We obtain labels from NHANES opioid prescription and self-report records (Appendix A). To classify individuals, models must integrate information about their diet, health conditions, and socioeconomic factors. Distinguishing active users from those in recovery is particularly hard, as both groups share similar risk factors and differ only in minor dietary and behavioral patterns. For all predictive models in this task, we remove User–Opioid_Level edges and never expose opioid_level nodes or features during training and inference, to avoid any label leakage.

**Evaluation Metrics.** In our dataset, the three classes are highly imbalanced, with at-risk users being relatively rare. Therefore, we use metrics that evaluate performance across all classes rather than favoring the majority class. We report four metrics: macro-averaged F1 (F1-macro), one-vs-rest AUC averaged across classes, geometric mean of per-class recalls (GMean), and overall accuracy. F1-macro gives equal weight to each class to assess balanced performance. GMean is particularly sensitive to how well the model identifies minority classes and reflects the balance between detecting at-risk and avoiding bias toward the majority. Accuracy and AUC offer additional perspectives on overall correctness and ranking quality.

**Baseline Models and Setup.** We evaluate representative methods using two data split configurations that simulate different training scenarios: 60/20/20 and 70/15/15 for train/validation/test sets. For each split, we randomly divide users and run experiments with multiple seeds, then report the mean and standard deviation of results. We organize baseline methods into three categories (Table 1). (i) Non-graph baseline: a multi-layer perceptron (MLP) operating on tabular user features, which quantifies the benefit of explicitly modeling interactions. (ii) Graph baselines: homogeneous GNNs (GCN (Kipf & Welling, 2017), GraphSAGE (Hamilton et al., 2018), GAT (Veličković et al., 2018)) trained on a simplified homogeneous

Table 1: Opioid misuse detection over GLEN Nutrition-Health Graph with two train-validation-test splits (60/20/20 and 70/15/15). LLM methods use LLaMA-3.1-8B, Qwen3-8B, and DeepSeek-R1-Distill-Qwen-7B.

| Methods | Train 60% − Valid 20% − Test 20% | | | | Train 70% − Valid 15% − Test 15% | | | |
|---|---|---|---|---|---|---|---|---|
| | F1-macro | AUC | GMean | Accuracy | F1-macro | AUC | GMean | Accuracy |
| MLP | 11.42±1.09 | 56.08±0.36 | 49.17±0.08 | 17.38±0.32 | 12.75±1.03 | 56.95±0.22 | 52.58±0.11 | 18.85±0.32 |
| GCN | 12.52±0.94 | 57.01±0.31 | 52.39±0.13 | 18.37±2.88 | 13.49±1.07 | 58.41±0.36 | 53.46±0.09 | 20.07±0.31 |
| GraphSAGE | 16.96±1.05 | 55.63±0.15 | 51.51±0.12 | 28.57±0.28 | 18.75±0.54 | 59.62±0.14 | 56.39±0.23 | 32.41±1.02 |
| GAT | 17.84±0.89 | 56.70±0.25 | 54.15±0.19 | 33.04±2.59 | 19.60±0.97 | 61.52±0.28 | 57.19±0.22 | 33.82±0.29 |
| RGCN | 19.22±0.51 | 58.85±0.34 | 54.31±0.24 | 33.13±0.77 | 20.56±0.13 | 56.91±0.04 | 50.86±0.03 | 37.49±0.22 |
| HGT | 21.63±0.74 | 57.42±0.35 | 50.80±0.39 | 39.79±0.23 | 20.80±0.89 | 56.48±0.32 | 52.41±0.27 | 37.70±0.27 |
| HAN | 25.85±0.06 | 70.58±0.09 | **65.08±0.12** | 47.43±0.19 | 24.25±0.09 | 71.23±0.05 | **65.67±0.10** | 43.40±0.24 |
| LLaMA-3.1 | 22.94±0.07 | 67.99±0.19 | 60.33±0.19 | 42.24±0.20 | 21.61±0.04 | 68.21±0.15 | 59.27±0.12 | 38.97±0.12 |
| Qwen3 | 23.82±0.07 | 67.90±0.17 | 60.89±0.18 | 44.73±0.29 | 22.91±0.08 | 69.76±0.07 | 60.73±0.06 | 42.84±0.27 |
| DeepSeek-R1 | 24.86±0.03 | 64.31±0.17 | 56.29±0.16 | 48.51±0.12 | 24.11±0.11 | 69.05±0.11 | 60.42±0.06 | 45.74±0.37 |
| LLaMA-3.1 + Graph | 27.99±0.14 | 70.28±0.33 | 61.61±0.23 | 54.73±0.31 | 29.28±0.10 | 69.74±0.29 | 60.29±0.38 | 58.63±0.56 |
| Qwen3 + Graph | 28.55±0.34 | 70.57±0.37 | 61.34±0.24 | 55.91±0.76 | 29.55±0.43 | 71.31±0.42 | 63.39±0.32 | 59.15±1.04 |
| DeepSeek-R1 + Graph | **31.45±0.15** | **70.74±0.37** | 60.43±0.20 | **67.20±0.49** | **30.22±0.21** | **71.97±0.33** | 63.29±0.28 | **60.82±0.66** |

graph that ignores node/edge types (or concatenates type features), enabling us to isolate the gain from connectivity alone; and heterogeneous GNNs (RGCN (Schlichtkrull et al., 2017), HGT (Hu et al., 2020), HAN (Wang et al., 2021b)) that preserve multi-relational structure via type-specific parameters. (iii) LLM baselines: large language models (LLaMA-3.1, Qwen3, DeepSeek-R1) applied to a serialized representation of each user's profile information (without graph neighborhood). We further evaluate *LLM + Graph* hybrids (e.g., DeepSeek-R1 + Graph) by injecting graph-derived neighborhood context into prompts as structured evidence. Table 1 reports F1-macro, AUC, GMean, and Accuracy for all baselines under these settings.

**Results.** Table 1 shows that graph structure is essential for identifying at-risk users whose patterns of misuse are often latent or "hidden". Compared to non-graph MLP baseline, message passing models such as GCN, GraphSAGE, and GAT significantly improve macro-F1 and GMean under both splits. This shows that node features alone are not enough. The graph structure captures complex relationships and risk patterns that cannot be detected from individual data points. Heterogeneous modeling further amplifies these gains. HAN attains the best GMean among classical GNNs in the 70/15/15 split, indicating that separating relation types (e.g., user-to-health vs. user-to-habit) reduces feature interference and improves sensitivity toward minority classes. LLMs perform well on average, but they struggle with class balance in highly imbalanced datasets. While their AUC values are competitive, their macro F1 and GMean scores lag behind the best graph models. This suggests that LLMs are susceptible to semantic bias from majority classes, and semantic reasoning alone cannot capture the fine-grained relational cues needed for minority class detection. The hybrid approach addresses this gap. DeepSeek-R1-Distill-Qwen-7B model with graph achieves the best results across metrics in both splits. The improvement over text-only LLMs suggests that graph structure acts as a guide, helping the LLM focus on relevant neighborhood patterns and make more stable predictions for rare users.

The results across the two dataset splits show a consistent trend. Adjusting the split ratio from 60/20/20 to 70/15/15 increased the number of training samples, but the relative ranking of the models remained stable. Although the accuracy of DeepSeek-R1 + Graph adjusts from 67.20±0.49 to 60.82±0.66, its continued dominance confirms that the model captures invariant structural properties rather than relying on specific overfitting. While pure LLMs improve primarily on accuracy and AUC with flatter macro F1 scores, the consistent gains of hybrid approaches across all metrics suggest that graph structure helps extract deeper value from additional training data. These results suggest a clear strategy: use graphs to organize clinical information and let LLMs analyze the resulting patterns. Graphs capture relationships that are hard to describe in text, while LLMs provide flexible reasoning across different user profiles.

Despite these improvements, the best-performing model achieves only 31.45% F1-macro, which highlights that three-class opioid risk detection under combined nutritional and socioeconomic signals remains an open problem. Table 1 also reveals a consistent decoupling between AUC and GMean. Several models rank users reasonably well by AUC yet fail to classify minority groups correctly, as reflected in low GMean scores.

Table 2: Personalized food recommendation over GLEN Nutrition-Health Graph. Results are reported as mean ± std%. The best performance is bolded and runner-ups are underlined.

| Methods | GNN | Graph CF | Contrastive | Specialized | Recall@20 | NDCG@20 | H-Score@20 | PA@20 | AvgTags@20 |
|---|---|---|---|---|---|---|---|---|---|
| GCN | ✓ | – | – | – | 9.65±0.27 | 7.68±0.16 | 26.45±1.17 | 14.41±0.54 | 6.64±0.39 |
| GraphSAGE | ✓ | – | – | – | 9.75±0.80 | 6.74±0.32 | 30.27±1.17 | 15.14±0.33 | 6.46±0.19 |
| GAT | ✓ | – | – | – | 10.17±0.30 | 8.58±0.17 | 32.24±0.30 | 16.39±0.38 | 6.90±0.32 |
| NGCF | – | ✓ | – | – | 12.93±0.05 | 7.98±0.07 | 33.98±1.11 | 17.62±0.14 | 7.09±0.40 |
| LightGCN | – | ✓ | – | – | 11.99±0.41 | 7.67±0.28 | 33.86±0.31 | 17.35±0.33 | 6.81±0.20 |
| SimGCL | – | ✓ | ✓ | – | 12.79±0.39 | 8.06±0.24 | 34.11±0.34 | 16.85±0.26 | 6.74±0.15 |
| SGL | – | ✓ | ✓ | – | 11.02±0.15 | 6.80±0.09 | 33.34±0.09 | 15.95±0.11 | 6.73±0.67 |
| LightGCL | – | ✓ | ✓ | – | 11.48±0.24 | 7.60±0.18 | 33.07±0.21 | 16.17±0.20 | 6.71±0.14 |
| RecipeRec | – | – | – | ✓ | 12.72±0.28 | 9.17±0.75 | 39.64±0.20 | 26.55±0.63 | 6.78±0.93 |
| HFRS-DA | – | – | – | ✓ | 12.78±0.27 | 9.20±0.75 | 34.21±0.17 | **27.78±0.75** | **7.30±0.72** |
| MOPI-HFRS | – | – | – | ✓ | **13.25±0.34** | **9.97±0.61** | **38.18±0.74** | 26.84±1.07 | 7.21±0.73 |

For screening tasks where identifying at-risk individuals is the primary goal, GMean is therefore a more informative metric than AUC alone. Among LLM-based methods, text-only models show competitive AUC but substantially lower F1-macro and GMean. This suggests that language models tend to favor the majority class when the label distribution is skewed, and that semantic reasoning alone is not sufficient for reliable minority class detection in clinical risk screening.

**Dietary and Clinical Signatures of Opioid Misuse.** To better understand what drives model performance, we analyze how medical conditions and dietary habits differ across the three user groups in NHANES. Table 7 summarizes these distributions and reports Chi-square test results. Every medical condition we examined, including sleep disorder, depression, obesity, hypertension, and CKD, shows a highly significant difference across groups ($p < 0.001$). Dietary habits follow the same pattern. Opioid users are more likely to report poor diet quality, high salt use, frequent consumption of frozen or ready-to-eat foods, and low supplement intake. These signals are spread across many behavioral and clinical dimensions, which is why graph-based methods recover them more effectively than feature-only baselines. The patterns also point to concrete targets for nutrition-informed intervention design.

## 4.2 Personalized Food Recommendation

Personalized food recommendation is central to nutrition-aware intervention because it translates risk screening into actionable daily choices. In practice, recommendations must be not only preference-aligned but also clinically appropriate under condition-specific constraints (e.g., sodium restriction for hypertension) and feasible under socioeconomic limitations such as affordability and food access. These interconnected needs constitute a multi-objective scenario that requires balancing relevance, health compliance, and feasibility, rather than simply optimizing user interaction. Therefore, we treat personalized food recommendation as a core task to evaluate whether the model can leverage the GLEN Nutrition-Health Graph to generate clinically and economically valuable recommendations.

**Task Definition.** This task is formulated as a top-$K$ food recommendation problem where, for each user in GLEN Nutrition-Health Graph, the model must rank candidate foods and return a list of $K$ items that are nutritionally appropriate and economically feasible, while still matching the user's preferences. Unlike conventional collaborative filtering that relies primarily on user-item interaction patterns, our formulation leverages the rich heterogeneous structure of GLEN Nutrition-Health Graph. User nodes connect not only to consumed foods but also to health condition nodes encoding clinical diagnoses (e.g., hypertension, diabetes), nutrition tag nodes specifying dietary requirements (e.g., low_sodium, high_fiber), and socioeconomic indicator nodes capturing financial constraints (poverty status, food insecurity). Food nodes link to ingredient and category nodes describing compositional structure, nutrition tags reflecting nutrient profiles, and price tags denoting affordability tiers. A successful recommender must therefore generate foods that align with user preferences while respecting health-derived nutritional constraints (e.g., restricting sodium for hypertensive users) and accommodating economic realities imposed by poverty and limited food access.

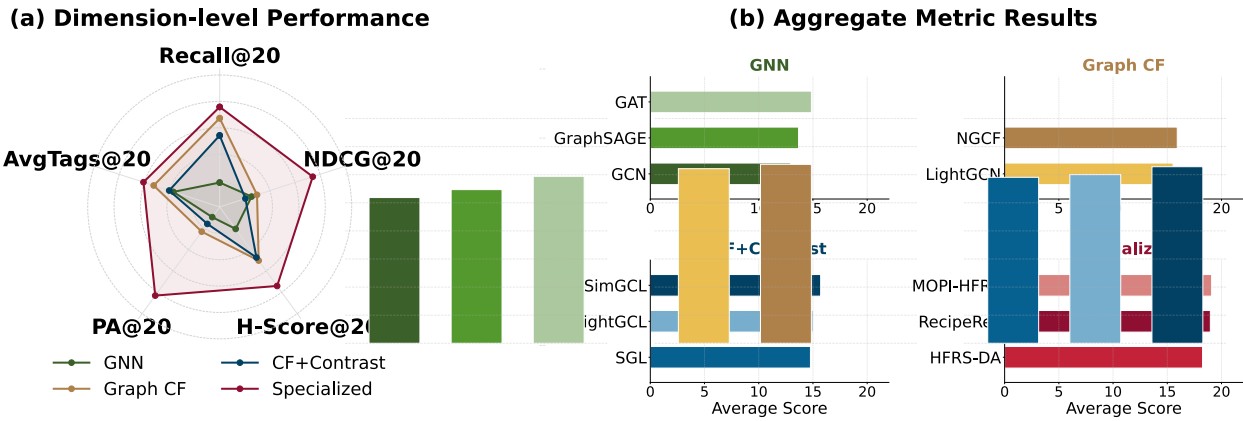

Figure 4: Personalized food recommendation performance across four model families (GNNs, collaborative filtering methods, contrastive-enhanced CF, and domain-specialized models) on GLEN-Bench.

**Evaluation Metrics.** We fix the recommendation list length to $K = 20$ and evaluate ranking quality using two standard metrics, Recall@20 and NDCG@20 (Normalized Discounted Cumulative Gain). To assess whether recommendations are consistent with user-specific dietary needs, we use the health-aware H-Score metric from prior work on personalized food recommendation: H-Score@20 measures, for each user, the proportion of recommended foods that share at least one required health or nutrition tag with the user, and then averages this value across users. To capture economic accessibility, we introduce PA@20 (Poverty Awareness@20). For each user, PA@20 computes the proportion of recommended foods whose price tags are compatible with that user's financial tags (for example low-price foods for users tagged with low income or food insecurity) and averages this proportion across users. Higher PA@20 indicates that the model adapts recommendations to the user's budget constraints. Finally, we report AvgTags@20, the average number of distinct relevant nutrition or health tags covered by the top-20 list, which reflects the informational richness and tag-level diversity of the recommended menu.

**Baseline Models and Setup.** Baselines for this task have three categories. (i) Classical graph neural network baselines: GCN (Kipf & Welling, 2017), GraphSAGE (Hamilton et al., 2018), and GAT (Veličković et al., 2018), applied to the user-food interaction graph extracted from GLEN Nutrition-Health Graph. (ii) SOTA graph-based recommendation baselines: SGL (Yu et al., 2022), LightGCN (He et al., 2020), SimGCL (Wu et al., 2021), LightGCL (Cai et al., 2023), and NGCF (Wang et al., 2019), which are widely used in general recommendation benchmarks and optimized for ranking metrics. (iii) SOTA food recommendation baselines: RecipeRec (Tian et al., 2022a), HFRS-DA (Forouzandeh et al., 2024), and MOPI-HFRS (Zhang et al., 2025b), which explicitly incorporate nutrition or health constraints and support multi-objective recommendation. All methods are trained and evaluated on the same user-food splits. We repeat experiments across multiple random seeds and report mean and standard deviation.

**Results.** Table 2 and Figure 4 link model behavior to architectural traits. Figure 4 summarizes these effects at the family level, with panel (a) reporting dimension-wise performance and panel (b) aggregating metric averages. Baseline GNNs (GCN/GraphSAGE/GAT) leverage local neighborhood aggregation on the user–food graph. GAT achieves 10.17±0.30 Recall@20 and 8.58±0.17 NDCG@20, indicating that neighborhood structure supports preference aligned ranking. However, without explicitly modeling health and price, improvements on H-Score@20 and PA@20 are limited. This indicates that interaction patterns alone cannot guarantee clinically and economically viable recommendations. Collaborative filtering variants further strengthen collaborative signals. NGCF improves ranking quality (e.g., 12.93±0.05 Recall@20 and 7.98±0.07 NDCG@20), while LightGCN simplifies propagation and SimGCL/SGL introduce contrastive regularization. These design choices yield steady gains on Recall and NDCG, yet H-Score and PA improve only marginally, reflecting the preference-centered objective. Constraint-aware models exhibit a different pattern. HFRS-DA encodes typed relations among users, foods, health conditions, nutrition tags, and prices, and applies relation-specific attention, achieving 34.21±0.17 H-Score@20 and 27.78±0.75 PA@20 while maintaining

12.78±0.27 Recall@20. This indicates that explicitly modeling relational constraints can improve compliance without sacrificing ranking quality. MOPI-HFRS further couples relevance and constraint objectives via prompt-driven multi-objective optimization. It reaches 13.25±0.34 Recall@20, 9.97±0.61 NDCG@20, 38.18±0.74 H-Score@20, and 26.84±1.07 PA@20, with 7.21±0.73 AvgTags@20, indicating broader coverage of nutrition- and health-related attributes.

These results demonstrate that ranking performance is decoupled from constraint satisfaction. Incorporating multi-relational health signals and constraint-aware objectives enables more feasible recommendations without sacrificing standard metrics like Recall and NDCG. This gap highlights a key finding of GLEN-Bench: existing recommendation architectures lack effective mechanisms for enforcing socioeconomic constraints, and PA@20 is the metric that makes this visible. Figure 4 further shows that GNN-based models consistently underperform specialized models on H-Score across all configurations, even when their Recall and NDCG scores are comparable. This suggests that preference signals and health constraint signals interfere with each other during graph message passing, and that optimizing for user-food interaction patterns does not naturally lead to clinically appropriate recommendations. Constraint-aware objectives and relation-specific modeling are therefore necessary components rather than optional additions for nutrition-aware recommendation.

### 4.3 Nutritional Question Answering

Nutritional question answering helps build trust in dietary recommendations. Patients and providers need clear explanations before changing diets, particularly in clinical contexts. Even when a model can rank foods well, real-world adoption depends on whether it can explain why a food is suitable (or unsuitable) given the user's health conditions, dietary requirements, and socioeconomic constraints. This task therefore complements risk detection and recommendation by evaluating evidence-grounded reasoning and interpretability. By requiring both tag-level attribution and language-based explanations grounded in the GLEN Nutrition-Health Graph, nutritional QA assesses whether models can produce faithful, personalized rationales rather than generic nutrition statements, supporting transparency, user trust, and informed intervention decisions.

**Task Definition.** Nutritional Question Answering (QA) tests whether models can reason over the GLEN Nutrition-Health Graph and generate accurate, interpretable answers to personalized nutrition questions. Each question presents a user, a candidate food, and relevant graph context, asking whether the food is suitable and which nutrients or tags support the decision. Following the task formulation in NGQA (Zhang et al., 2024a), we adopt two coupled views: Multi-Label Classification (-ML) requires models to identify nutritional tags and determine their alignment or contradiction with a user's health profile, thereby evaluating the ability to reason over nuanced relationships in structured graph data. Text Generation (-TG) focuses on producing natural-language justifications for food healthiness to assess the model's capacity for interpretable and user-friendly reasoning in personalized dietary contexts.

**Evaluation Metrics.** We evaluate QA along two corresponding output views. For multi-label classification (ML), we report Accuracy, Recall, Precision, F1, and AUC to measure how well the model recovers the ground-truth supporting nutrition tags, balancing coverage (Recall) against spurious predictions (Precision/F1). For text generation (TG), we use ROUGE-1/2/L, BLEU, and BERTScore to compare generated explanations with references in terms of overlap and semantic similarity, reflecting fluency and content fidelity.

**Baseline Models and Setup.** We benchmark a range of LLMs and graph enhanced prompting strategies. For each backbone model (LLaMA3 and GPT-4), we first test a plain setting that directly prompts the model with the question, without requiring explicit reasoning steps. We then include several reasoning oriented prompting methods such as CoT-Zero (Kojima et al., 2022) and CoT-BAG (Wang et al., 2023), which encourage chain of thought reasoning, and ToT (Yao et al., 2023) and GoT (Besta et al., 2024), which organize intermediate thoughts in a tree or graph structure. To examine how graph knowledge affects performance, we test methods that combine retrieval from the GLEN Nutrition-Health Graph with LLM reasoning. We evaluate four Graph RAG frameworks: KAPING (Baek et al., 2023) and ToG (Sun et al., 2023) retrieve and simplify subgraphs before generating answers, G-retriever (He et al., 2024) focuses on retrieving high-quality graph information for reasoning, and KAR (Xia et al., 2025) incorporates retrieved knowledge triples and refines answers iteratively. All methods share the same question templates and are

Table 3: Nutritional question answering over GLEN Nutrition-Health Graph

| Model | Method | Multi-label Classification (-ML) | | | | | Text Generation (-TG) | | | | |
|---|---|---|---|---|---|---|---|---|---|---|---|
| | | Accuracy | Recall | Precision | F1 | AUC | ROUGE-1 | ROUGE-2 | ROUGE-L | BLEU | BERT |
| LLaMA3 | Plain | 0.3274 | 0.9880 | 0.3248 | 0.4863 | 0.7336 | 0.6088 | 0.5531 | 0.5997 | 0.3626 | 0.9436 |
| | CoT-Zero | 0.3282 | **0.9896** | 0.3262 | 0.4882 | 0.7349 | 0.6118 | 0.5566 | 0.6027 | 0.3649 | 0.9435 |
| | CoT-BAG | 0.3496 | 0.9335 | 0.3856 | 0.5292 | 0.7701 | 0.6743 | 0.5971 | 0.6569 | 0.4213 | 0.9516 |
| | ToT | 0.3153 | 0.7376 | 0.3832 | 0.4787 | 0.7076 | 0.6408 | 0.5678 | 0.6232 | 0.3844 | 0.9436 |
| | GoT | 0.3117 | 0.9566 | 0.3135 | 0.4709 | 0.7043 | 0.6122 | 0.5487 | 0.6045 | 0.3547 | 0.9422 |
| | KAPING | 0.3177 | 0.9876 | 0.3191 | 0.4803 | 0.7163 | 0.5931 | 0.5286 | 0.5840 | 0.3704 | 0.9416 |
| | KAR | 0.3275 | 0.9216 | 0.3648 | 0.5102 | 0.7625 | 0.6250 | 0.5589 | 0.6141 | 0.3677 | 0.9444 |
| | ToG | **0.3640** | 0.7556 | 0.4082 | 0.5118 | 0.7359 | **0.7359** | **0.6697** | **0.7180** | **0.5086** | **0.9591** |
| | G-retriever | 0.3558 | 0.7811 | **0.5054** | **0.5723** | **0.7948** | 0.6493 | 0.5763 | 0.6330 | 0.3907 | 0.9462 |
| GPT4 | Plain | 0.3089 | **0.9938** | 0.3096 | 0.4701 | 0.7168 | 0.6080 | 0.5242 | 0.5944 | 0.3489 | 0.9405 |
| | CoT-Zero | 0.3114 | 0.9912 | 0.3124 | 0.4727 | 0.7216 | 0.6227 | 0.5425 | 0.6090 | 0.3627 | 0.9430 |
| | CoT-BAG | **0.3213** | 0.9880 | 0.3250 | 0.4856 | 0.7383 | 0.5307 | 0.4172 | 0.5168 | 0.2563 | 0.9194 |
| | ToT | 0.2568 | 0.8158 | 0.3171 | 0.4474 | 0.6915 | 0.6651 | 0.5904 | 0.6514 | 0.4135 | **0.9457** |
| | GoT | 0.3088 | 0.9619 | 0.3129 | 0.4705 | 0.7076 | 0.6261 | 0.5476 | 0.6123 | 0.3674 | 0.9390 |
| | KAPING | 0.3038 | 0.9903 | 0.3065 | 0.4658 | 0.7149 | 0.5906 | 0.5024 | 0.5783 | 0.3318 | 0.9371 |
| | KAR | 0.3117 | 0.9357 | 0.3481 | **0.4951** | **0.7512** | 0.6120 | 0.5227 | 0.5982 | 0.3499 | 0.9392 |
| | ToG | 0.2894 | 0.5780 | **0.3934** | 0.4220 | 0.6663 | 0.6002 | 0.4818 | 0.5853 | 0.3356 | 0.9347 |
| | G-retriever | 0.3210 | 0.7882 | 0.3808 | 0.4923 | 0.7371 | **0.6830** | **0.6087** | **0.6686** | **0.4551** | 0.9455 |

evaluated on both ML and TG metrics, which allows us to compare pure prompting against graph aware pipelines and to quantify how much structured nutritional knowledge improves personalized nutritional QA.

**Results.** Table 3 shows a clear pattern across backbones. Plain prompting reasons over the input without structured evidence, so it tends to over predict supporting tags and generates rationales that contain generic phrases. Adding chain of thought variants such as CoT-Zero and CoT-BAG, or search style controllers such as ToT and GoT, organizes intermediate steps but still operates without external nutritional structure. The result is small and inconsistent gains in F1 and AUC and only modest movement in ROUGE and BERTScore. Reasoning tokens help with step ordering yet do not supply the fine grained tag level facts the questions require.

Graph-based retrieval methods perform much better because they focus on finding high-quality evidence. KAPING and ToG first retrieve and filter relevant subgraphs, then use them for generation. This improves AUC and ROUGE compared to direct prompting because the model now has access to specific nutrient and health information. G-retriever goes further by emphasizing retrieval accuracy. With LLaMA3, it achieves the best multi-label scores in its group, with the highest F1 and AUC, and also improves ROUGE and BERTScore. This shows that better subgraphs lead to fewer incorrect tags and more accurate explanations. KAR uses a different approach by incorporating retrieved knowledge triples during answer generation. With GPT-4, this method achieves the best F1 score among GPT-4 variants, while G-retriever produces the best ROUGE-1, ROUGE-L, and BERTScore. This demonstrates that strong retrieval improves explanation quality even with powerful language models.

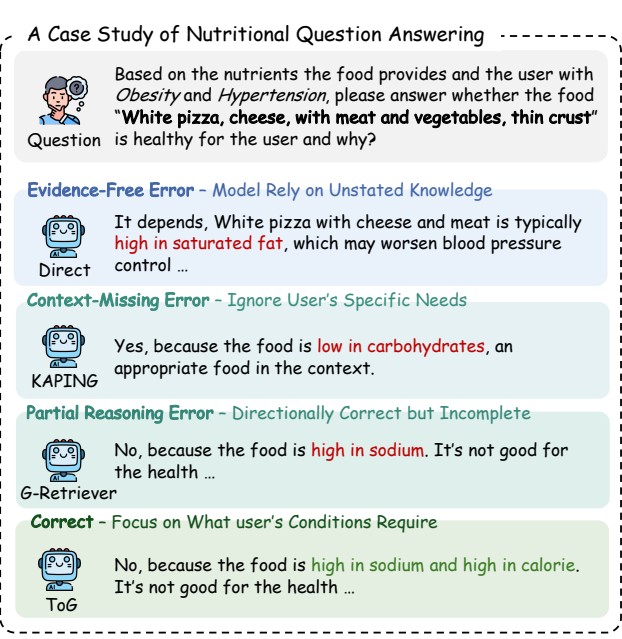

Figure 5: A case study of error analysis across various model architectures.

Two general patterns emerge from these results. First, retrieval quality determines multi-label per-

formance: accurate graph information leads to better AUC and F1 scores and fewer incorrect tags. Second, model scale does not uniformly improve text generation quality. With the same retrieval method, LLaMA3 achieves comparable or higher BERTScore than GPT-4 in most settings, and LLaMA3 with ToG produces the highest ROUGE and BERTScore across both models. GPT-4 shows advantages primarily in certain retrieval-free settings (e.g., ToT). This suggests that retrieval quality may be a more decisive factor than model scale for nutritional QA, and that smaller models paired with effective graph retrieval can match or surpass larger models. These findings suggest choosing the retrieval method before selecting the language model. Use methods like KAR when tag accuracy matters most, and use high-quality retrievers like G-retriever when clear, well-supported explanations are the priority.

We also note that overall classification accuracy across all models remains around 0.35, which reflects the difficulty of multi-label tag prediction in this setting. Correctly identifying which nutrition tags apply to a food requires precise numerical reasoning over nutrient values, a capability that pure language models do not reliably possess without structured evidence. This points to an inherent limitation of prompt-based approaches for nutrition-aware QA.

## 5 Conclusion

While prior nutrition-aware benchmarks remain fragmented and do not support integrated evaluation under real-world constraints, we present GLEN-Bench to advance nutritional health research by combining diet, clinical data, and socioeconomic factors in a single benchmark. By unifying these signals in one heterogeneous graph, GLEN-Bench enables researchers to study nutrition-aware decision making in a setting that more closely resembles real clinical workflows. Our evaluation shows that integrating graph-based reasoning with language models improves performance on risk detection, personalized recommendation, and interpretable explanations. The gains are consistent across tasks, which suggests that structured evidence and language-based reasoning play complementary roles rather than substitute ones. This approach better reflects real-world intervention workflows than treating these tasks separately. Results further highlight the importance of explicit relational structure and constraint-aware modeling for learning meaningful representations and generating clinically appropriate, economically feasible recommendations. We also find that ranking quality alone does not guarantee clinical or economic suitability, which underscores the value of evaluating multiple dimensions together. Beyond opioid use disorder, GLEN-Bench generalizes to diverse nutrition-sensitive conditions such as diabetes, cardiovascular disease, and chronic kidney disease, and supports broader tasks such as meal planning, adherence prediction, and food and drug interaction analysis. The schema is easy to extend, since new conditions can be added by attaching condition-specific nutrition tags without changing the overall graph design. By standardizing data, tasks, and evaluation protocols, GLEN-Bench enables systematic, reproducible progress toward fair and accessible nutrition-aware AI.

Several open challenges emerge from our results. The low PA@20 scores across all recommendation models suggest that enforcing socioeconomic constraints during training remains an unsolved problem, and future work could explore constraint-aware objective functions or post-hoc reranking strategies. It would also be valuable to study how to balance affordability with user preference, since overly strict price filtering may reduce the diversity of recommended foods. The persistent gap between AUC and GMean in risk detection points to the need for imbalance-aware training methods tailored to multi-relational graph settings. This is especially important for screening tasks, where missing an at-risk user is more costly than a false alarm. On the QA side, evaluating the faithfulness of generated explanations with respect to retrieved graph evidence is an important direction, as fluent rationales may not always reflect the underlying evidence. Stronger faithfulness checks would help build trust among clinicians and patients, who often rely on the reasoning behind a recommendation rather than the recommendation itself. More broadly, extending GLEN-Bench to longitudinal settings and incorporating dynamic pricing and local food availability signals would better reflect real-world intervention contexts. Adding richer social and behavioral signals, such as cultural preferences and household structure, would further close the gap between benchmark evaluation and practical nutrition care. We hope GLEN-Bench can serve as a long-term platform that supports continued progress on trustworthy and accessible AI for personalized nutrition.

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

# A GRAPH CONSTRUCTION DETAILS

## A.1 Data Sources and Preprocessing

The GLEN Nutrition-Health Graph is constructed by integrating three population-scale resources to jointly model *clinical status*, *dietary intake*, and *socioeconomic affordability*. Together, these sources provide a unified foundation for studying how health conditions, dietary behaviors, food composition, and real-world access constraints interact at scale.

***NHANES (2003–2020).*** We use NHANES cycles from 2003 to 2020 and extract participant-level information from: (i) *Demographics* (e.g., age, sex, race/ethnicity), (ii) *Examination* and *Laboratory* sections (e.g., BMI, blood pressure, metabolic biomarkers), (iii) *Questionnaire* modules for diagnosed medical conditions (e.g., obesity, hypertension, depression) and behavioral factors (e.g., salt use, supplement intake), and (iv) *Dietary* modules for 24-hour dietary recalls and food consumption logs. We treat each participant as a **user** node and consolidate all available attributes into a unified user profile via ID-based joins across modules. In addition, we derive socioeconomic attributes including annual household income and food insecurity indicators, which enable annotating users with poverty status and food insecurity risk—introducing real-world feasibility constraints that are rarely modeled in nutrition-health datasets. When multiple records exist across modules, we keep a single unified user record; missing values are preserved as blank (or imputed with simple statistics depending on the downstream model setting).

We acknowledge that aggregating NHANES cycles from 2003 to 2020 into a static graph introduces temporal heterogeneity. However, NHANES maintains consistent survey methodology across cycles, and our graph models cross-sectional associations rather than temporal trends. Temporal calibration (e.g., inflation adjustment) is an important direction for future work.

***FNDDS and WWEIA.*** Each NHANES consumed food item is recorded with a food code. We map NHANES food codes to the Food and Nutrient Database for Dietary Studies (FNDDS) to obtain comprehensive nutrient profiles, and to What We Eat In America (WWEIA) to obtain standardized hierarchical food categories. This mapping allows us to create **food** nodes equipped with nutrient vectors, and to organize foods through **category** nodes and structured **ingredient** composition. Based on standardized nutrient thresholds, we further derive interpretable **nutrition_tag** labels to support downstream tasks that require explicit nutritional justifications.

***USDA Purchase-to-Plate.*** To incorporate real-world affordability constraints, we augment each food with price estimates from the USDA Purchase-to-Plate pipeline. Prices are normalized to *cost per 100 grams* and discretized into a small set of affordability tiers, represented as **price_tag** nodes (Low/Medium/High). These tiers approximate the relative economic burden of selecting specific foods and enable models to jointly reason about nutritional suitability and accessibility, which distinguishes GLEN-Bench from prior nutrition-oriented graph resources that typically ignore economic feasibility.

## A.2 Graph Schema and Statistics

We model GLEN Nutrition-Health Graph as a heterogeneous graph $G = (\mathcal{V}, \mathcal{E}, \mathcal{T}, \mathcal{R})$, where each node $v \in \mathcal{V}$ is assigned a node type $\tau(v) \in \mathcal{T}$ and each edge $e \in \mathcal{E}$ is assigned a relation type $\phi(e) \in \mathcal{R}$. The graph includes user, food, ingredient, category, dietary habit, health condition, nutrition tag, price tag, poverty condition, and opioid-level nodes, with edges capturing observed consumption, food composition/category structure, nutrition and price tagging, and user-specific clinical/socioeconomic associations. We report the detailed node/relation statistics in Table 4.

| Item | Count |
|---|---|
| Nodes | # User = 104,244, # Food = 9,640, # Ingredient = 36,591, # Category = 36,718, # Habit = 48, # Health Condition = 19, # Nutrition Tag = 18, # Poverty Condition = 1, # Price Tag = 3, # Opioid Level = 3 |
| Relations | # User–Food = 1,803,215, # User–Habit = 652,277, # User–Health Condition = 145,650, # User–Poverty Condition = 35,033, # User–Opioid Level = 98,448, # Food–Ingredient = 31,510, # Food–Category = 8,388, # Food–Nutrition Tag = 52,582, # Food–Price Tag = 7,683, # Health Condition–Nutrition Tag = 23, # Poverty Condition–Price Tag = 1 |

Table 4: The statistics of GLEN Nutrition-Health Graph.

## A.3 Node Features and Representations

We associate each node with a feature vector, with different sources depending on the node type.

***User Nodes.*** Each user node $u$ is associated with a feature vector $x_u \in \mathbb{R}^{d_u}$, constructed by concatenating demographic attributes and clinical/laboratory measurements extracted from NHANES (e.g., BMI, blood pressure, metabolic biomarkers). Diagnosed conditions, dietary habits, poverty-related constraints, and opioid status are modeled as explicit neighbors rather than being fully absorbed into $x_u$.

***Food Nodes.*** Each food node $f$ is associated with a continuous nutrient vector $x_f \in \mathbb{R}^{d_f}$, derived from FNDDS nutrient profiles. These nutrient values are also used to derive threshold-based nutrition tags (Table 5).

***Categorical/semantic nodes.*** For non-numeric node types such as ingredient, category, habit, health_condition, nutrition_tag, price_tag, and poverty_condition, we represent their semantics using pre-trained BERT embeddings over their textual names or descriptions, producing fixed-length vectors as node representations. This provides a unified embedding space for symbolic concepts that do not have natural numeric features.

## A.4 Dietary Habit Extraction

Dietary habit information is compiled from NHANES diet- and behavior-related questionnaires. Due to the categorical diversity of these responses, we perform a structured feature-to-habit mapping: a team of four reviewers identifies questionnaire indicators that describe habitual dietary patterns (e.g., awareness toward healthy eating, frequency of consuming certain food types). For each habit-related feature, we apply a **top/bottom quantile strategy**: participants in the top 10% and bottom 10% (by frequency or ordinal response) are assigned corresponding habit tags. This process yields **48** distinct habit concepts, each modeled as a **habit** node, connected to users via User–Habit edges.

## A.5 Health Risk Label Generation (Opioid Status)

We construct a three-level opioid status label for evaluation and create **opioid_level** nodes (3 levels). Users are connected to their corresponding level via User-Opioid_Level edges. Following NHANES drug-use and prescription-related records, we operationalize:

- **Active Opioid Users:** users with evidence of heroin use within the past year, or continuous prescription opioid use for over 90 days.

- **Opioid-Recovered Users:** users with a history of opioid use but without the criteria of active misuse in the current period (derived from NHANES history/current-use signals).

- **Normal Users:** users with no record indicating opioid misuse or long-term prescription history.

Note that not all users have opioid-related records; therefore, we only create User-Opioid_Level edges when the relevant modules are available (98,448 labeled users in the graph statistics).

## A.6 Tagging Scheme

GLEN includes two coupled tagging mechanisms: (i) **food-level nutrition tags** derived from nutrient thresholds, and (ii) **health-driven dietary requirement tags** that connect clinical indicators to recommended nutrition constraints.

**Nutrition Tags from Nutrient Thresholds (Food → Tag)** We define a fixed set of **nutrition_tag** nodes and assign them to foods using nutrient thresholding. Specifically, for each food $f$ and nutrient dimension $k$, we compare the nutrient value against predefined low/high thresholds to determine whether $f$ satisfies a tag. The thresholds used in GLEN are summarized in Table 5, together with recommended reference values (NRV) when applicable.

**Tag assignment.** Given food nutrient vector $x_f$, a tag is added if the corresponding threshold condition holds. For example:

- **Low Sodium**: sodium$(f) \leq 120$ mg;

- **High Protein**: protein$(f) \geq 15$ g;

- **Low Sugar**: sugar$(f) \leq 5$ g;

- **High Fiber**: fiber$(f) \geq 6$ g.

We then create Food-NutritionTag edges for all satisfied tags (52,582 edges in the final graph).

**Health Indicators to Dietary Requirement Tags (Condition → Tag)** In addition to food-level tags, we encode clinical dietary requirements by connecting health_condition nodes to recommended nutrition_tag nodes. This captures domain-guided associations such as "hypertension → low sodium" and enables downstream tasks (e.g., personalized recommendation and QA) to reason over interpretable constraint paths.

| Nutrients | Low Threshold | High Threshold | NRV |
|---|---|---|---|
| Calories (kcal) | 40 | 225 | 2000 |
| Carbohydrates (g) | 55 | 75 | - |
| Protein (g) | 10 | 15 | 50 |
| Saturated Fat (g) | 1.5 | 5 | 20 |
| Cholesterol (mg) | 20 | 40 | 300 |
| Sugar (g) | 5 | 22.5 | - |
| Dietary Fiber (g) | 3 | 6 | - |
| Sodium (mg) | 120 | 200 | 2000 |
| Potassium (mg) | 0 | 525 | 3500 |
| Phosphorus (mg) | 0 | 105 | 700 |
| Iron (mg) | 0 | 3.3 | 22 |
| Calcium (mg) | 0 | 150 | 1000 |
| Folic Acid ($\mu$g) | 0 | 60 | 400 |
| Vitamin C (mg) | 0 | 15 | 100 |
| Vitamin D ($\mu$g) | 0 | 2.25 | 15 |
| Vitamin B12 ($\mu$g) | 0 | 0.36 | 2.4 |

Table 5: Nutrient thresholds and recommended values for nutrition tags.

We summarize the condition-to-tag mapping used in GLEN in Table 6. Each row defines a set of recommended tags for a health indicator. We instantiate these as Health_Condition-Nutrition_Tag edges (23 edges in total).

| Health Indicator | Associated Tags |
|---|---|
| Obesity | Low Calorie; High Fiber |
| Diabetes | Low Sugar; Low Carb; High Fiber |
| Anemia | High Iron; High Vitamin C; High Folate Acid; High Vitamin $B_{12}$ |
| Chronic kidney disease (CKD) | Low Protein; Low Sodium; Low Phosphorus; Low Potassium |
| Dyslipidemia | Low Saturated Fat; Low Cholesterol; High Fiber |
| Hyperuricemia | Low Purine |
| Sleep disorder | High Vitamin D |
| Depression | High Folate Acid; High Vitamin D |
| Liver disease | Low Sodium |
| Weight loss/Low calorie diet | Low Calorie |
| Low fat/Low cholesterol diet | Low Cholesterol; Low Saturated Fat; High Fiber |
| Low salt/Low sodium diet | Low Sodium |
| Sugar free/Low sugar diet | Low Sugar |
| Diabetic diet | Low Sugar; Low Carb; High Fiber |
| Weight gain/Muscle building diet | High Calorie; High Protein |
| Low carbohydrate diet | Low Carb |
| High protein diet | High Protein |
| Renal/Kidney diet | Low Protein; Low Sodium; Low Phosphorus; Low Potassium |

Table 6: Health indicators and associated nutrition tags in GLEN Nutrition-Health Graph.

**Price Tags and Affordability Constraints (Food → PriceTag)** We discretize normalized price (cost per 100g) into three tiers and represent them with price_tag nodes: low_price, medium_price and high_price.

We discretize prices into three tiers because (i) the USDA estimates are aggregate national averages where fine-grained continuous modeling would be misleading, and (ii) discrete tiers align with how food assistance programs categorize accessibility. Finer-grained affordability modeling is a direction for future work.

**Poverty Condition and Economic Access (User → PovertyCondition)** We represent extreme socioeconomic constraint with a poverty_condition node, and connect users meeting the criterion through User-

PovertyCondition edges. We further anchor affordability constraints by linking PovertyCondition–PriceTag, enabling models to jointly reason about nutritional suitability and economic feasibility.

## B   EXPERIMENTAL SETTINGS

### B.1   Environmental Settings

All experiments are conducted on a Linux server equipped with two NVIDIA A100 GPUs (40GB each). We implement all methods in Python 3.10.18 with PyTorch 2.4.0 and PyTorch Geometric 2.7.0. We use standard scientific computing libraries (NumPy, SciPy, scikit-learn) for preprocessing and evaluation, and HuggingFace Transformers (v4.57.1) for LLM-based baselines. When applicable, we enable mixed-precision training/inference to improve throughput on A100 GPUs.

### B.2   Data Splits and Reproducibility

**Train/valid/test splits.**   For opioid misuse detection (Section 4.1), we evaluate under two split configurations: **60/20/20** and **70/15/15** for train/validation/test, split at the **user** level to prevent leakage. For personalized food recommendation (Section 4.2), we use a single **60/20/20** split under the same user-level partitioning protocol.

The different split configurations reflect task-specific needs: two splits for risk detection to test sensitivity under class imbalance, and a single split for recommendation where ranking stability is the primary concern.

**Repeated runs.**   For each split configuration, we evaluate each method using a fixed set of random seeds and report the mean and standard deviation across runs. For training-based methods, we select the checkpoint with the best validation performance and report test results using that checkpoint.

### B.3   Hyperparameters

We adopt unified training settings across methods whenever applicable to ensure fair comparison, and select model checkpoints based on validation performance.

**Opioid misuse detection.**   All GNN-based models are optimized with Adam for 500 epochs using hidden dimension 256 and dropout 0.6. We report mean±std over 10 random seeds. We apply dropout in message passing layers and choose learning rate and weight decay from a small preset range on the validation set.

**Personalized food recommendation.**   We formulate recommendation as implicit-feedback ranking and train all recommenders using the BPR objective with negative sampling. We fix the recommendation list length to $K=20$. For a balanced comparison across methods, we use a unified configuration with hidden dimension 128 and Adam optimizer (learning rate 1e-3, L2 regularization 1e-6) for up to 500 epochs, and apply a learning-rate decay step every 200 epochs. All methods are trained and evaluated on the same 60/20/20 user-level split (performed once with a fixed split seed), and we select the best validation checkpoint for final test reporting. If a baseline is unstable under the shared configuration, we apply minimal adjustments within the same hyperparameter family to ensure convergence, while keeping the protocol and evaluation unchanged.

**Nutritional question answering (multi-label + explanation).**   For QA baselines (prompting, reasoning-oriented prompting, and graph-grounded pipelines), we evaluate both multi-label tag prediction and explanation generation following Section 4.3. To reduce variance, we keep decoding settings fixed within each LLM backbone and do not tune decoding hyperparameters on the test set.

Table 7: Statistical analysis of medical status and dietary habits across opioid, recovered, and normal user groups.

| | Category | # Opioid Users | # Recov. Users | # Normal Users | p-value |
|---|---|---|---|---|---|
| | **NHANES Data** | 3104 | 437 | 94907 | - |
| **Medical Status** | Sleep Disorder | 989 | 99 | 7607 | < 0.001 |
| | Depression | 1337 | 167 | 10733 | < 0.001 |
| | Obesity | 1155 | 138 | 17554 | < 0.001 |
| | Hypertension | 667 | 80 | 10038 | < 0.001 |
| | Diabetes | 313 | 23 | 3681 | < 0.001 |
| | Anemia | 749 | 69 | 20606 | < 0.001 |
| | CKD | 589 | 48 | 5286 | < 0.001 |
| | Dyslipidemia | 1328 | 188 | 26867 | < 0.001 |
| | Hyperuricemia | 692 | 80 | 10570 | < 0.001 |
| | Liver disease | 480 | 122 | 13267 | < 0.001 |
| **Dietary Habits** | Drinks little or no milk | 591 | 77 | 11609 | < 0.001 |
| | Adds lots of salt at table | 507 | 96 | 9331 | < 0.001 |
| | Eats lots of frozen food | 296 | 49 | 7022 | < 0.001 |
| | Eats often outside the home | 266 | 65 | 9318 | < 0.001 |
| | Eats many ready to eat meals | 366 | 57 | 9271 | < 0.001 |
| | Takes few or no supplements | 1217 | 222 | 52799 | < 0.001 |
| | Uses lots of salt in preparation | 968 | 185 | 33868 | < 0.001 |
| | Claims to have a poor diet | 1090 | 149 | 15975 | < 0.001 |
| | Light cigarette smoker | 529 | 65 | 11443 | < 0.001 |
| | Heavy cigarette smoker | 172 | 57 | 3193 | < 0.001 |
| | Uses tobacco rarely | 90 | 21 | 1765 | < 0.001 |
| | Uses tobacco often | 407 | 65 | 2771 | < 0.001 |
| | Drinks Alcohol more than average | 923 | 200 | 7799 | < 0.001 |

*Note: p*-values are calculated using Chi-square tests to indicate statistical significance across the three groups.

## C  FINE-GRAINED CORRELATION BETWEEN DIETARY PATTERNS AND OPIOID MISUSE

To contextualize the opioid misuse detection task (Section 4.1), we analyze fine-grained differences in medical status and dietary/lifestyle habits across opioid, recovered, and normal user groups in NHANES. Table 7 summarizes the number of users exhibiting each indicator in the three groups and reports significance using Chi-square tests.

As shown in Table 7, we observe consistent and statistically significant distribution shifts across both medical conditions (e.g., sleep disorder, depression, obesity, hypertension, diabetes, CKD, dyslipidemia, hyperuricemia, liver disease) and dietary habits (e.g., diet quality, frozen/ready-to-eat food consumption, eating outside the home, supplement intake, salt-related behaviors), together with tobacco- and alcohol-related indicators. These results indicate that opioid misuse is associated with systematic behavioral and clinical signatures beyond any single variable. This motivates Task 1 to evaluate whether computational models can screen at-risk users by jointly leveraging multi-view signals encoded in the GLEN Nutrition-Health Graph.

## D  PROMPT DESIGN

For nutritional question answering (Section 4.3), we use a unified prompt template with method-specific wrappers. Each instance provides (i) a user and a candidate food, (ii) a compact graph context serialized as a node list and an edge list, and (iii) an explicit output format constraint to ensure parseable predictions.

We control output strictness through difficulty-conditioned notes. For medium difficulty, the model outputs a comma-separated list of nutrition tags (e.g., high_carb, low_sodium) chosen from a fixed option set. For hard difficulty, the model outputs a binary decision (Yes/No) followed by a short rationale that explicitly references the selected tags. Across prompting baselines, we keep the core instruction fixed and only vary

the wrapper that introduces the auxiliary evidence (e.g., plain prompting, chain-of-thought style instruction, or graph-grounded retrieval outputs such as KAPING/ToG/GoT/G-Retriever/KAR).

While the core template remains identical, different baselines introduce the auxiliary evidence with lightweight wrapper instructions that mirror their intended reasoning or retrieval mechanism. **Plain** uses only the shared instruction and the serialized graph context without additional reasoning cues. **CoT-Zero** appends a step-by-step instruction, encouraging the model to explicitly (i) extract the food's nutrition tags from the graph, (ii) compare them against the user's health conditions and required tags, and (iii) form a final judgement. **CoT-BaG** follows the same core objective but emphasizes that the input evidence is a directed graph description, prompting the model to interpret nodes and edges as structured facts.

For retrieval-augmented graph baselines, the wrapper primarily clarifies the provenance and intended use of the provided evidence. **KAPING** and **ToG** are framed as graph-grounded pipelines that supply a relevant subgraph/context block to be used as the sole evidence for answering. **ToT** and **GoT** additionally provide intermediate reasoning artifacts produced by tree/graph-structured search and aggregation, and the wrapper instructs the model to use these artifacts as supporting evidence rather than generating unsupported facts. **G-Retriever** specifies that the context is a compact subgraph selected by Prize-Collecting Steiner Tree optimization to balance relevance and brevity, encouraging faithful use of the retrieved neighborhood. Finally, **KAR** introduces knowledge-aware retrieved information (e.g., filtered triples and refined context) and instructs the model to ground both tag attribution and explanation in these retrieved facts.

Across all wrappers, we keep the allowed nutrition-tag vocabulary fixed and enforce the same output-format constraints, so that performance differences reflect the quality of evidence selection and reasoning rather than prompt leakage or formatting freedom.

# E  ADDITIONAL RELATED WORK

**Graph Representation Learning** Graph neural networks have become essential for learning from structured relational data. Heterogeneous graph neural networks (Zhang et al., 2019) extend standard GNNs to handle multiple node and edge types, enabling richer modeling of complex systems with diverse entities and relationships. This multi-relational capability is particularly valuable when reasoning requires understanding interactions across different types of connections. Real-world graphs often contain noise, missing edges, and structural errors. Graph structure refinement methods (Zhao et al., 2023) address these challenges by learning to adaptively adjust topology during training, improving robustness to data quality issues. Many graph learning tasks also face severe class imbalance, where target nodes or patterns are rare. Contrastive learning approaches (Qian et al., 2022; 2025) have shown effectiveness in such settings through self-supervised objectives that learn representations without requiring extensive labels, enabling pre-training on large unlabeled graphs before fine-tuning on specific tasks. Fairness in graph learning has gained attention as models may amplify biases present in training data, leading to disparate performance across subgroups (Liu et al., 2023). Recent work on graph foundation models (Wang et al., 2024b) aims to build transferable representations that generalize across different graphs and domains, reducing the need for task-specific training data and enabling efficient adaptation to new applications.

# F  ETHICS AND PRIVACY STATEMENT

GLEN-Bench is constructed from publicly available, population-level resources and is designed with privacy and ethical considerations as first-class requirements. Our primary human data source, the National Health and Nutrition Examination Survey (NHANES), is released under strict confidentiality safeguards and public-data policies. The NHANES records we use are de-identified and do not contain personally identifiable information (PII) such as social security numbers, names, phone numbers, or physical addresses. We additionally follow a data minimization principle: our benchmark focuses on variables necessary for nutrition-health assessment (e.g., demographic attributes, clinical measurements, dietary recalls, and questionnaire-derived behaviors) and does not attempt to re-identify individuals or link records to external identity sources.

---

### Prompt 1: Nutritional QA Prompt Template

**System Message:** You are a nutrition-aware QA agent. Use **only** the provided graph evidence. Follow the output format strictly.

**Difficulty Note:**

- **Medium:** output nutrition tags only, as a comma-separated list.

- **Hard:** output `Yes/No` + a short evidence-grounded rationale that explicitly mentions the tags.

**Graph Evidence (Serialized):**

- **Node List:** `[(id, {type, attr}), ...]`

- **Edge List:** `[(src, rel, dst), ...]`

**Question:**

- **Decision:** Is the candidate food healthy for the user? (`Yes/No`)

- **Attribution:** Which nutrition tags support the decision?

- **Explanation (Hard only):** Provide a short rationale grounded in the given evidence.

**Output Format:**

- **Medium:** `low_carb, low_sodium, low_protein, ...`

- **Hard:** `Yes, because the food is low_sodium, low_carb, ...`

---

Figure 6: Prompt template for nutritional question answering. The prompt combines a unified system instruction, difficulty-conditioned output constraints, and a serialized graph evidence block to encourage evidence-grounded decisions and explanations.

GLEN-Bench integrates NHANES with food composition and category information (FNDDS/WWEIA) and aggregated price/access signals (USDA Purchase-to-Plate) to study clinically appropriate and feasible nutrition interventions. These auxiliary resources do not introduce individual-level identifiers; they provide food-level nutrition and affordability context that supports constraint-aware modeling. We report results in aggregate and do not release any information that could enable tracing predictions back to specific survey participants.

We recognize that nutrition and substance-use–related variables can be sensitive. As a benchmark, GLEN-Bench is intended for research on equitable decision support rather than automated clinical decision-making. Any real-world deployment should include additional safeguards, including access control, secure storage, audit logging, and human oversight, and should treat recommendations and explanations as protected health information within clinical workflows. By operating within established survey policies and emphasizing de-identification, minimization, and responsible use, we aim to uphold strong ethical integrity and privacy protection throughout dataset construction, evaluation, and dissemination.

## G  LIMITATIONS AND DISCUSSION

GLEN-Bench provides a unified setting for end-to-end, constraint-aware nutritional health assessment, but several limitations remain. First, the benchmark is built on observational survey data (NHANES) and derived resources, which introduces confounding and measurement noise (e.g., imperfect 24-hour recalls

and questionnaire responses). As a result, strong performance should be interpreted as capturing reliable associations rather than causal effects, and models may exploit spurious shortcuts. A promising direction is to incorporate evaluations that stress-test robustness to confounding and missingness, such as sensitivity analyses, subgroup robustness checks, and counterfactual or intervention-style variants (e.g., whether a model's recommendation changes appropriately when specific constraints or nutrient targets are perturbed).

Second, the feasibility signals in GLEN-Bench are necessarily simplified. Price tags and access-related constraints are estimated at an aggregate level and may not reflect local availability, seasonal price variation, or individual purchasing context. Similarly, practical dietary decision-making often depends on additional constraints not fully represented here—cultural preferences, allergies, cooking skills, time budget, household composition, and condition- or medication-specific restrictions. Future versions of the benchmark could enrich these dimensions by adding finer-grained SDoH variables, dynamic affordability/availability indicators, and structured preference profiles, enabling more realistic evaluation of what it means for recommendations to be simultaneously clinically appropriate and actionable.

Third, while our task suite captures a realistic workflow (screening → recommending → explaining), the current metrics do not fully measure clinical utility, long-term adherence, or safety beyond tag-level compliance. In particular, LLM-based methods can generate fluent rationales that are not faithfully grounded in graph evidence. An important future direction is to strengthen faithfulness and safety evaluation: auditing whether explanations cite retrieved evidence, checking consistency between predicted tags and generated rationales, calibrating model confidence (including abstention on uncertain cases), and incorporating human-in-the-loop evaluation protocols that reflect clinical and public-health decision processes. More broadly, GLEN-Bench opens opportunities for additional tasks such as nutrient-aware substitution, meal planning under budget and nutrient constraints, adherence prediction, and food–drug interaction analysis, which would further align benchmarking with real-world nutrition interventions.

Overall, we view GLEN-Bench as an evolving platform. The current release establishes a reproducible, multi-task benchmark with explicit socioeconomic constraints, while future work can deepen causal rigor, broaden population coverage, enrich constraint modeling, and improve faithfulness and safety assessment for nutrition-aware decision support.

