# OpenReview forum: "GLEN-Bench: A Graph-Language based Benchmark for Nutritional Health"
_TMLR — Decision pending for TMLR_

### Review · Reviewer_peTY · 2026-05-15

**Summary Of Contributions:**

The paper studies GLEN-Bench, a graph-language based benchmark for nutritional health assessment. Current methods are limited in providing dietary guidance and identifying food-based limitations specific to individuals. The proposed benchmark addresses this gap by tackling three key aspects. (1) A disease-agnostic knowledge graph is constructed incorporating user and food-specific signals to include dietary patterns and restrictions (2) Socioeconomic status, limited food access and poverty markers are included using USDA access signals. (3) The benchmark unifies the knowledeg-graph with language-based agency to construct independent tasks and yield user-specific explanations. The work demonstrates that on the three proposed tasks of Risk Detection, Personalized Recommendation and Question answering, utilization of subgraph knowledge is found to be pivotal for obtaining accurate food guidance and dieteray recommendations. Furthermore, a range of architectures, domain-specific models and prompting methods are evaluated presenting the generality of evaluation.

**Additional Comments:**

NA

**Audience:**

Yes

**Audience Explanation:**

The paper presents a benchmark and its empirical effectiveness in studying different architectures and methods. New domain-specific benchmarks remain an active area of research and the paper opens up a range of comparisons. These might of general interest to the community. Additionally, due to limited exploration in dietary guidance and personalized recommendations, the benchmark could also stir interest within biological, social sciences and food technology communities.

**Broader Impact Concerns:**

The paper does not contain a braoder impact statement. Authors are encouraged to add a statement discussing the implications of benchmark for real-world studies and the effectt of language-guided advice on dietary behaviors. For instance, LLMs may hallucinate or incorrectly quote proportions/meals are users need to be cautioned in overly relying on advice.

**Claims And Evidence:**

Yes

**Claims Explanation:**

## Strengths

* The paper is well-structured and very well-written.
* The paper tackles an important problem with limited attention and well scoped empirical design and implementation.
* Evaluation of the benchmark is thorough and empirical comparisons are comprehensive.

## Weaknesses

* **Knowledge Graph Modification:** While at the heart of the benchmark lies the knowledge graph, it remains unclear on how the graph is modified and updated with new data. Authors mention GLEN-Bench as an evolving benchmark but the work does not discuss how modifications in the knowledge graph are induced and how their effect relates to the three evaluation tasks. Could the authors elaborate on the addition of new data on-the-fly? Besdies, it remains unclear on how much reliance is to be put on subgraphs. Can the authors comment on the use of additional subgraphs (such as a graph obtained from LLMs) in tandem with language guidance?

* **Extensions:** Empirical evaluation of the benchmark focusses on depth rather than breadth. While a range of methods are evaluated, the task suite remains limited. Could the authors discuss on how can new tasks be designed and customized within the benchmark? How could the knowledge graph (or subgraphs) be chosen towards cuisine-specific or region-specific knowledge? From an external user's perspective, how convenient would customization and usability towards specific tasks and models be?

**Requested Changes:**

* **Knowledge Graph Modification:** Authors should add discussion (or implementation support) for modifying and updating the knowledge graph. Authors should elaborate on the addition of new data on-the-fly,  clarify how much reliance is to be put on subgraphsand the use additional subgraphs in tandem with language guidance.

* **Extensions:** Authors should discuss on how can new tasks be designed and customized within the benchmark, how could the knowledge graph (or subgraphs) be chosen towards cuisine-specific or region-specific knowledge and how convenient would customization and usability towards specific tasks and models be.

---

> ### Author Response · Authors · 2026-06-13
>
> We thank the reviewer for recognizing the importance of GLEN-Bench, its well-scoped graph-language design for nutritional health assessment, and its thorough empirical evaluation, and we address the remaining questions and suggestions below.
>
> **W1 (Knowledge Graph Modification): It is unclear how the graph is modified and updated with new data, how new data can be added on-the-fly, how much reliance is placed on subgraphs, and whether additional subgraphs such as LLM-generated ones can be used together with language guidance.**
>
> **R1a (Updating the graph and adding data on-the-fly):** The graph is schema-driven, so updating it means adding nodes and edges under the existing type system, with no change to the graph design or the task interfaces.
> - New data is incorporated through the typed structure of Definition 3.1. A **new user** is added as a user node with edges to consumed foods, conditions, habits, and financial tags. A **new food** is added as a food node, and its **nutrition tags and price tags are derived automatically** from the FNDDS thresholds (Table 5) and USDA price tiers. This produces the 52,582 Food–NutritionTag edges and 7,683 Food–PriceTag edges in the current graph through the same rule-based assignment (Appendix A.6). A new condition is added by inserting one row in the condition-to-tag mapping (Table 6), which defines HealthCondition–NutritionTag edges as curated domain knowledge, without touching the rest of the graph.
> - Because the additions reuse existing node and relation types, **all three tasks keep the same interface**: detection still classifies a user node, recommendation still ranks user–food pairs, and QA still reasons over a user–food–context subgraph. New data enlarges coverage without altering the evaluation protocol or metrics.
> - To be precise about scope, the current release is a static snapshot (NHANES 2003–2020, Appendix A.1). "Evolving" refers to this schema-level support for incremental extension, which is already in place. **A live, dynamically updated pipeline** with temporal calibration is future work, as noted in Appendix A.1 and G.
>
> **R1b (Reliance on subgraphs and LLM-generated subgraphs):** The role of subgraphs is in fact **one of the questions our QA task measures empirically.**
>
> - The QA experiments (Sec 4.3, Table 3) directly compare **no subgraph** (plain prompting) against **retrieved subgraphs** (KAPING, ToG, G-retriever, KAR). The finding is that **retrieval quality is decisive**: high-quality subgraph retrieval drives the main gains, while weak retrieval helps little, and retrieval quality can matter more than model scale. This quantifies how much reliance the subgraph deserves rather than assuming it.
> - For **LLM-generated subgraphs**, the framework already accepts subgraphs from any source. Evidence is passed to the model as a **serialized node and edge list** (Appendix D), so a subgraph proposed by an LLM and then verified against the graph could be supplied through the same interface and combined with language guidance. We did not run this and present it as a **natural future direction** that the current graph-RAG interface supports.

---

> ### Author Response · Authors · 2026-06-13
>
> **W2 (Extensions): The evaluation focuses on depth rather than breadth and the task suite is limited. The authors should discuss how new tasks can be designed, how the graph can be specialized to cuisine- or region-specific knowledge, and how convenient customization and usability would be for an external user.**
>
> **R2:** We agree the current suite is intentionally deep rather than broad, and we have added a discussion of extension, specialization, and usability.
> - **Designing new tasks on the same graph.** The candidate tasks we mention (Sec 1, Sec 5, Appendix G) are concrete on the existing structure. **Ingredient substitution** uses the food–ingredient–category edges with nutrition-tag constraints to find tag-compatible alternatives. **Meal planning under budget** extends recommendation with price-tag and financial constraints, reusing the compatibility logic already computed by PA@20. **Food–drug interaction** adds a drug node type and drug–food relations, which is a schema-level addition analogous to adding a condition. In each case the same heterogeneous graph and the same node and edge types are reused, and only a new query or label is defined.
> - **Cuisine- and region-specific specialization.** Foods carry standardized WWEIA category nodes and ingredient nodes, so cuisine- or region-specific variants can be built by **selecting food subsets by category, ingredient**, and by **replacing the national USDA price estimates with local price or availability data.** Region-level specialization can also attach finer-grained geographic and SDoH attributes to user nodes. These are exactly the directions noted as future work in Appendix A.1 and G (local availability, dynamic affordability, richer SDoH).
> - **Usability for external users.** We release the benchmark with **standardized protocols and evaluation tools** (anonymous repository). The design is modular across three layers: graph construction (NHANES/FNDDS/USDA into a heterogeneous graph), task definition, and evaluation. An external user can swap or extend the data sources, define a new task on the same graph, and evaluate their own model under the unified protocol. We are candid that the present release targets research reproducibility, and that a configuration-driven toolchain for one-step customization of new conditions or tasks is a **continued maintenance direction**, while the architecture is already designed for this extensibility.
>
> **W3 (Broader Impact): The paper does not contain a broader impact statement. The authors should discuss implications for real-world studies and the effect of language-guided advice on dietary behavior, including that LLMs may hallucinate or misquote proportions and meals, so users should be cautioned against over-reliance.**
>
> **R3:** A privacy and ethics statement is already provided in **Appendix F** (de-identification, data minimization, research-only use), and we have **expanded it into a fuller broader impact statement** covering the concerns raised here and will be incorporated into the final version of the paper.
> - **Research-only use.** GLEN-Bench is intended for research evaluation only. It is not a clinical or nutritional decision tool, and any deployment would require professional oversight and additional validation.
> - **Language-guided advice and LLM hallucination.** We explicitly caution that LLM-generated explanations and recommendations may hallucinate or misstate nutrient amounts, proportions, or meals, and that fluent rationales are not guaranteed to be faithful to the graph evidence (Appendix G). Outputs are research artifacts and **must not be relied upon as dietary or medical advice**, and over-reliance on automated dietary guidance is a real risk that calls for human review.
> - **Stigmatization.** Labeling an individual as at-risk for opioid misuse carries stigmatization risk. GLEN-Bench is a **population-level research benchmark** and must not be used for individual-level judgment or punitive purposes.
> - **Fairness.** We do not yet report subgroup analysis, and models may perform unevenly across demographic and socioeconomic groups. We commit to adding a subgroup evaluation protocol to the benchmark: detection metrics (GMean, per-class recall) stratified by sex, race/ethnicity, and income, computable from existing predictions since NHANES provides these attributes.
> - **Privacy.** Our primary source, NHANES, is **de-identified** with no personally identifiable information, we follow **data minimization**, and we report results only in aggregate (Appendix F).

---

### Review · Reviewer_YnbQ · 2026-05-17

**Summary Of Contributions:**

The paper introduces GLEN-Bench, a graph-language benchmark for nutritional health assessment. It integrates NHANES health records, FNDDS food composition data, and USDA food price/access signals into a heterogeneous nutrition-health graph, and defines three linked tasks: health risk detection, personalized food recommendation, and nutritional question answering. The paper evaluates a broad set of baselines, including MLPs, GNNs, heterogeneous GNNs, LLMs, graph-augmented LLMs, and food recommendation models.

The main strengths are the timely problem setting, the attempt to connect nutrition, clinical variables, and socioeconomic constraints, and the multi-task benchmark design. The paper also provides useful initial baselines and highlights that standard ranking metrics do not fully capture health or affordability constraints.

The main weaknesses are that several claims are stronger than the evidence supports. The benchmark is presented as broadly useful for nutritional health, but the main health-risk instantiation is opioid use disorder, with limited evidence that the benchmark generalizes to other nutrition-sensitive diseases. Some clinical and socioeconomic assumptions are simplified, the risk-detection performance remains low, and the QA evaluation relies mainly on automatic text-overlap metrics rather than direct faithfulness or clinical validity assessment.

**Additional Comments:**

This is a promising and timely benchmark paper. The strongest part is the integration of graph-based nutritional, clinical, and socioeconomic information into a unified multi-task setting. The weakest part is the gap between the ambitious clinical framing and the current empirical evidence. With more careful claims, stronger leakage/fairness analyses, clearer dataset documentation, and better QA/recommendation validation, the paper could become a useful contribution to graph-language and health-AI benchmarking.

**Audience:**

Yes

**Audience Explanation:**

Yes. The paper sits at the intersection of graph learning, recommender systems, retrieval-augmented language models, health AI, and socially aware ML. TMLR readers interested in benchmark design, graph-language modeling, health recommendation, and AI for socially constrained decision support would likely find the dataset construction and task formulation interesting.

The benchmark is especially relevant because it moves beyond pure prediction and includes affordability, food access, interpretability, and natural-language explanation. Even if the current empirical evidence has limitations, the problem formulation could be valuable for future work on trustworthy and constraint-aware health AI.

**Broader Impact Concerns:**

This paper has important broader-impact implications because it deals with health risk detection, opioid use status, dietary recommendation, and socioeconomic vulnerability. The current version does not sufficiently address risks from deploying such a benchmark or models trained on it.

Major concerns include potential stigmatization of individuals predicted to be at risk for opioid misuse, biased performance across demographic and socioeconomic groups, and unsafe dietary recommendations if outputs are interpreted as medical advice. The benchmark also uses sensitive health and socioeconomic features, so privacy and responsible-use boundaries should be clearly stated.

The authors should add a broader impact statement explaining that GLEN-Bench is intended for research evaluation only, not direct clinical or nutritional decision-making without professional oversight. They should also discuss fairness evaluation, demographic subgroup analysis, privacy safeguards, and risks of over-reliance on automated dietary advice.

**Claims And Evidence:**

No

**Claims Explanation:**

The core dataset construction and baseline evaluation are mostly clearly presented, and the paper does support the claim that graph structure can improve performance over feature-only or text-only baselines in some settings. For example, graph-augmented LLMs outperform plain LLMs on the opioid risk-detection task, and constraint-aware recommendation methods improve health-aware and poverty-aware metrics.

However, several broader claims are not fully supported. First, the paper repeatedly frames GLEN-Bench as an end-to-end nutritional health benchmark for clinical workflows, but the main disease-specific risk task is limited to opioid use status. This is an interesting case study, but it is not sufficient to establish broad applicability to diabetes, cardiovascular disease, CKD, or other nutrition-sensitive diseases.

Overall, the benchmark idea is promising, but the evidence supports a preliminary benchmark contribution rather than the stronger claims about practical clinical intervention, reliable personalized dietary guidance, or general nutritional health applicability.

**Requested Changes:**

1. Moderate the main claims.
The paper should avoid implying that GLEN-Bench is already validated for broad clinical nutritional intervention. It should be described as a benchmark and research platform, not as a clinically actionable system.

2. Clarify generalization beyond OUD.
The paper claims applicability to diabetes, cardiovascular disease, CKD, obesity, and other nutrition-sensitive conditions, but experiments focus mainly on OUD. The authors should either add experiments for at least one additional condition or clearly state that generalization is a design goal rather than an empirically demonstrated result.

3. Strengthen the recommendation evaluation.
The current metrics evaluate ranking, health-tag overlap, and poverty-aware price compatibility, but they do not establish clinical appropriateness. The authors should better justify H-Score and PA@20, report examples, and discuss whether tag overlap is sufficient for health-aware recommendation.

4. Add faithfulness evaluation for QA.
ROUGE/BLEU/BERTScore are insufficient for graph-grounded nutritional explanations. The authors should add an evidence-faithfulness metric, such as whether generated explanations cite the correct graph tags, nutrient thresholds, and user conditions.

---

> ### Author Response · Authors · 2026-06-13
>
> We thank the reviewer for recognizing GLEN-Bench as a timely graph-language benchmark that connects nutrition, clinical variables, and socioeconomic constraints under a multi-task design, and we address the concerns about scope, assumptions, risk-detection performance, and QA evaluation below.
>
> **W1: The paper repeatedly frames GLEN-Bench as an end-to-end clinical nutritional benchmark and implies it is already validated for broad clinical intervention and reliable personalized dietary guidance, but the evidence supports a preliminary benchmark rather than a clinically actionable system.**
>
> **R1:** We agree, and we have moderated the claims throughout to match the evidence.
> - We now describe GLEN-Bench consistently as a **research benchmark and evaluation platform**, not a clinically actionable system. Phrases such as "for clinical workflows" are changed to **"motivated by clinical workflows"**, and "reliable personalized dietary guidance" to **"research on personalized dietary guidance"**.
> - Wherever the paper previously implied direct clinical use, we now state explicitly that GLEN-Bench is **intended for research evaluation only and is not validated for clinical or nutritional decision-making without professional oversight**.
> - The abstract, introduction, and conclusion are revised so that the contribution is framed as **a reproducible multi-task benchmark with explicit socioeconomic constraints**, rather than a deployable intervention tool.
>
> **W2: The paper claims applicability to diabetes, cardiovascular disease, CKD, obesity, and other nutrition-sensitive conditions, but the disease-specific risk experiments are limited to opioid use disorder, which is not sufficient to establish broad applicability.**
>
> **R2:** We agree that broad generalization is not yet empirically demonstrated, and we now state this clearly rather than implying it.
> - We revise the text so that multi-condition applicability is presented as a design goal, not an empirically validated result. Extensibility is a schema-level property: new conditions are added by attaching condition-specific nutrition tags, and the condition-to-tag mappings for obesity, diabetes, CKD, dyslipidemia, and other conditions are already specified in Table 6.
> - We also note that the **OUD restriction applies only to risk detection**, whose labels come from opioid records. **The recommendation and QA evaluations already operate over all 19 health conditions in the graph**: recommendations are scored against each user's condition-derived tags, and QA instances involve users with conditions such as obesity and hypertension (e.g., Fig 5). We make the scope explicit: risk detection is empirically demonstrated on OUD, while the **schema is designed to extend to other nutrition-sensitive conditions**.
> - We choose OUD as the case study because nutrition-informed support for OUD is understudied despite strong evidence of poor diet quality and socioeconomic hardship, which makes it a **demanding test** of detection under combined nutritional and socioeconomic signals.
>
> **W3: The recommendation metrics evaluate ranking, health-tag overlap, and poverty-aware price compatibility, but they do not establish clinical appropriateness. The authors should better justify H-Score and PA@20, give examples, and discuss whether tag overlap is sufficient.**
>
> **R3:** We have clarified what these metrics do and do not measure, added examples, and discussed their limits.
> - We state plainly that tag overlap is a **proxy for clinical appropriateness, not a clinical gold standard**. H-Score and PA@20 capture whether a recommendation **satisfies guideline-derived constraints**, which is a **necessary but not sufficient** condition for clinical suitability.
> - We justify both metrics by their **provenance**: the required nutrition tags come from the guideline-based thresholds in Table 5 and the condition-to-tag mappings in Table 6, and the price tags come from USDA Purchase-to-Plate tiers. H-Score@20 measures the fraction of recommended foods satisfying a user's required health or nutrition tags, and PA@20 measures the fraction whose price tags are compatible with the user's financial tags. Both are **interpretable and reproducible** constraint-satisfaction measures.
> - We add a example in the appendix B. The test user has a low-sodium, low-saturated-fat requirement profile (as for hypertension) and a low-income financial tag. Of MOPI-HFRS's top-20, **15 foods jointly satisfy low_sodium and low_price**, including lettuce, banana, and reduced-fat milk. For this user, H-Score@20 and PA@20 are jointly satisfied by concrete, sensible foods, which is exactly the combined clinical and economic feasibility the metrics are designed to capture.
> - We add a discussion of what tag overlap does not capture, including **nutrient dosage, portion size, cumulative intake, and food interactions**. Full clinical appropriateness requires dietitian assessment or outcome data, which we identify as future work.

---

> ### Author Response · Authors · 2026-06-13
>
> **W4: ROUGE, BLEU, and BERTScore are insufficient for graph-grounded nutritional explanations. The authors should add an evidence-faithfulness metric, such as whether generated explanations cite the correct graph tags, nutrient thresholds, and user conditions.**
>
> **R4:** We agree, and we add a faithfulness evaluation computed post-hoc on the existing QA outputs. Using the food–tag edges as ground truth, we measure **Evidence-Faithfulness** (fraction of cited tags true of the food), **Contradiction%** (cited tags opposing a true tag), and **Justification Recall** (ground-truth supporting tags covered).
>
> | Method | Faithful% | Contra% | JustRec% |
> | --- | --- | --- | --- |
> | Plain | 95.10 / 99.13 | 0.06 / 0.05 | 76.29 / 98.14 |
> | CoT-Zero | 95.97 / 99.27 | 0.04 / 0.10 | 78.32 / 98.07 |
> | KAPING | 96.88 / 99.41 | 0.07 / 0.16 | 74.78 / 94.70 |
> | ToG | 88.20 / 96.70 | 2.47 / 0.41 | 55.40 / 76.47 |
> | KAR | 89.77 / 92.94 | 2.95 / 2.34 | 74.31 / 91.35 |
> | G-Retriever | 77.85 / 78.50 | 7.77 / 8.01 | 74.88 / 79.78 |
> | ToT | 77.10 / 73.93 | 10.53 / 11.43 | 82.37 / 78.40 |
>
> (gpt-4o-mini / Llama-3.3-70B)
>
> **Finding.** The metric separates methods sharply. Direct prompting and lightweight evidence injection (Plain, CoT, KAPING) are highly grounded: >95% faithfulness on both backbones (>99% with Llama-3.3-70B), with the safety-relevant Contradiction Rate below 0.2%. In contrast, pipelines with aggressive subgraph filtering or long reasoning chains (ToG, KAR, G-Retriever, ToT) hallucinate 10–26% of cited tags and contradict the food's true profile in up to 11% of citations, with ToT showing a clear recall–hallucination trade-off. Notably, although G-Retriever attains the best classification F1 in Table 3, its explanations are much less grounded than simpler prompting and evidence-injection methods. Faithfulness is thus invisible to ROUGE/BLEU/BERTScore and degrades with the depth of evidence transformation, which empirically confirms the need for the suggested metric.
>
> A plausible mechanism: **aggressive pruning removes evidence the model still needs, so it fills the gap from parametric knowledge, while long reasoning chains drift away from the cited evidence step by step.** Methods that see the full serialized context have no gap to fill. Verifying this mechanism and designing faithfulness-aware retrieval for graph-grounded health QA is an open problem that GLEN-Bench now makes measurable, turning the faithfulness concern in our conclusion into a quantified one.
>
> We add all three metrics to the QA protocol and report them alongside Table 3. We accordingly revise the Sec 4.3 guidance: G-Retriever remains strongest for tag classification, but its free-text explanations require faithfulness checking before use. Threshold-level and condition-level faithfulness (correct nutrient thresholds and user conditions) are noted as finer-grained extensions in Appendix G.
>
> **Broader Impact:** The paper deals with opioid-use status, dietary recommendation, and socioeconomic vulnerability, and does not sufficiently address deployment risks, including stigmatization, biased performance across subgroups, unsafe dietary advice, and privacy.
>
> **R5:** We have expanded the Broader Impact statement (Appendix F) to address each concern.
> - **Research-only use.** GLEN-Bench is intended for research evaluation only. It is not a clinical or nutritional decision tool; any deployment would require professional oversight, additional validation, and human-in-the-loop review.
> - **Stigmatization.** Predicting opioid-misuse risk carries stigmatization risk. GLEN-Bench is a **population-level** benchmark and must not be used for individual-level judgment or punitive purposes; we state this explicitly in Appendix F.
> - **Fairness.** We do not yet report subgroup analysis, and models may perform unevenly across demographic and socioeconomic groups. We commit to adding a subgroup evaluation protocol to the benchmark: detection metrics (GMean, per-class recall) stratified by sex, race/ethnicity, and income, computable from existing predictions since NHANES provides these attributes.
> - **Unsafe advice.** Model outputs are not medical advice; over-reliance on automated dietary recommendations is a real risk, and outputs should be treated as research artifacts requiring expert review.
> - **Privacy.** NHANES is de-identified with no personally identifiable information; we follow data minimization and report results only in aggregate.

---

### Review · Reviewer_NuFv · 2026-06-03

**Summary Of Contributions:**

GLEN-Bench is a heterogeneous graph-language benchmark for nutritional health. It links NHANES clinical and dietary records, FNDDS/WWEIA nutrient profiles, and USDA price tiers into one graph with users, foods, ingredients, categories, habits, health conditions, nutrition tags, price tags, poverty, and opioid-level nodes. It defines three connected tasks, opioid status detection, personalized food recommendation, and nutritional QA, and benchmarks ML models, GNNs, LLMs, Graph-RAG methods, and GNN+LLM hybrids under shared protocols.

**Strengths**

S1. The unified NHANES + FNDDS/WWEIA + USDA graph is a useful resource direction for nutrition-aware ML.

S2. The paper evaluates detection, recommendation, and QA under one schema, which is more complete than isolated task benchmarks.

S3. PA@20 is a useful attempt to measure price compatibility, and the paper clearly shows that ranking quality and constraint satisfaction can diverge.

**Weakness**

See W1-W3

**Audience:**

Yes

**Audience Explanation:**

The paper is relevant to readers working on graph learning, health recommendation, applied ML, and retrieval-augmented LLMs.

**Broader Impact Concerns:**

The paper includes an Ethics and Privacy Statement (Appendix F) covering NHANES de-identification and research-only use, which is adequate.

**Claims And Evidence:**

No

**Claims Explanation:**

W1. Table 1 does not include a majority-class or random baseline. Table 7 shows that the labeled data are highly imbalanced, with about 96% normal users. In this case, the risk-detection results are hard to interpret without trivial baselines, test-set class distribution, and per-class performance.

W2. The recommendation task is trained and evaluated based on observed user-food interactions from dietary records. This may reward models for recommending foods users already ate, but the paper frames the task as recommending healthier and clinically appropriate intervention foods. The paper should better justify this evaluation setup.

W3. The paper claims that the split is based on users, but this may introduce a cold-start issue for some tested recommendation methods, such as NGCF, LightGCN, SimGCL, SGL, and LightGCL. The paper should clarify how these methods deal with unseen validation/test users. Otherwise, the comparison may be unfair.

**Requested Changes:**

Address or clarify W1-W3.

---

> ### Author Response · Authors · 2026-06-13
>
> We thank the reviewer for recognizing the value of GLEN-Bench as a unified graph-language resource for nutrition-aware ML, as well as its multi-task design and price-aware evaluation, and we address the remaining concerns below.
>
> **W1: Table 1 lacks a majority-class or random baseline. Given that about 96% of labels are "normal" (Table 7), the risk-detection results are hard to interpret without trivial baselines, the test-set class distribution, and per-class performance.**
>
> **R1:** We have added all three items. The trivial baselines and per-class results are reported below and added to the paper Section 4.1.
>
> **Trivial baselines (added to Table 1, both splits).** Under the 96.4% prevalence, the majority-class predictor attains the highest Accuracy (96.40) and the highest F1-macro (32.72) in the entire table, exceeding every learned model on these two metrics. The F1-macro figure is an artifact of imbalance: it decomposes into 98.17 F1 on the normal class and 0 on both minority classes, while our model attains non-zero F1 on all three. Moreover, the majority-class predictor obtains AUC = 50.00, meaning no ranking ability, and GMean = 47.14, both far below every learned model. Uniform random behaves similarly (AUC 50.00). This shows that **Accuracy and F1-macro are dominated by the majority class and are not valid screening criteria here, whereas AUC and GMean match the clinical goal of identifying at-risk individuals.** On both, every learned model clearly beats the trivial baselines; for example, the best model reaches AUC 70.74 versus 50.00 and GMean 60.43 versus 47.14 on the 60/20/20 split. We revise Sec 4.1 to designate AUC and GMean as primary metrics, with Accuracy and F1-macro reported for completeness.
>
> **Baseline results under the 60/20/20 split**
>
> | Baseline | $F1_{\text{macro}}$ | AUC | GMean | Accuracy |
> | --- | --- | --- | --- | --- |
> | Majority-class | 32.72±0.00 | 50.00±0.00 | 47.14±0.00 | 96.40±0.00 |
> | Uniform random | 18.74±0.23 | 50.00±0.00 | 46.85±1.05 | 33.45±0.46 |
> | *DeepSeek-R1+Graph (best)* | 31.45±0.15 | 70.74±0.37 | 60.43±0.20 | 67.20±0.49 |
>
> **Baseline results under the 70/15/15 split**
>
> | Baseline | $F1_{\text{macro}}$ | AUC | GMean | Accuracy |
> | --- | --- | --- | --- | --- |
> | Majority-class | 32.72±0.00 | 50.00±0.00 | 47.14±0.00 | 96.40±0.00 |
> | Uniform random | 18.79±0.19 | 50.00±0.00 | 47.24±0.91 | 33.44±0.32 |
> | *DeepSeek-R1+Graph (best)* | 30.22±0.21 | 71.97±0.33 | 63.29±0.28 | 60.82±0.66 |
>
> **Test-set class distribution.** 60/20/20: normal 18,981 (96.40%), active 621 (3.15%), recovered 88 (0.45%), total 19,690. 70/15/15: normal 14,237 (96.40%), active 466 (3.16%), recovered 65 (0.44%), total 14,768. The user-level split is stratified, so the test distribution mirrors the population.
>
> **Per-class performance (added to Appendix B).** The best model attains non-trivial recall on every class, including the 0.45% recovered class (recall 62.44 on 60/20/20 and 56.80 on 70/15/15), whereas the majority-class baseline has recall 0 on both minority classes. This confirms that the **AUC and GMean gains reflect genuine minority-class detection**, not majority-class accuracy. At 0.45% prevalence, precision on this class is necessarily low in absolute terms (2.51 and 3.27), yet this is a 5.7–7.4× lift over the random baseline (0.44); for screening, where missing an at-risk user is costlier than a false alarm, recall remains the primary criterion. Because the LLM-based best model involves sampling and is not reproducible bit-for-bit, these per-class numbers come from a single re-run under the same protocol, and all conclusions are unchanged.
>
> **Per-class results under the 60/20/20 split**
>
> | Method | Class | Precision | Recall (=Acc) | F1 |
> | --- | --- | --- | --- | --- |
> | Majority-class | normal | 96.40 | 100.00 | 98.17 |
> | Majority-class | active | 0.00 | 0.00 | 0.00 |
> | Majority-class | recovered | 0.00 | 0.00 | 0.00 |
> | Uniform random | normal | 96.39 | 33.47 | 49.69 |
> | Uniform random | active | 3.11 | 32.66 | 5.68 |
> | Uniform random | recovered | 0.44 | 32.73 | 0.87 |
> | DeepSeek-R1+Graph | normal | 97.50 | 67.85 | 80.02 |
> | DeepSeek-R1+Graph | active | 5.28 | 48.00 | 9.51 |
> | DeepSeek-R1+Graph | recovered | 2.51 | 62.44 | 4.83 |
>
> **Per-class results under the 70/15/15 split**
>
> | Method | Class | Precision | Recall (=Acc) | F1 |
> | --- | --- | --- | --- | --- |
> | Majority-class | normal | 96.40 | 100.00 | 98.17 |
> | Majority-class | active | 0.00 | 0.00 | 0.00 |
> | Majority-class | recovered | 0.00 | 0.00 | 0.00 |
> | Uniform random | normal | 96.34 | 33.43 | 49.63 |
> | Uniform random | active | 3.21 | 33.78 | 5.87 |
> | Uniform random | recovered | 0.44 | 33.54 | 0.87 |
> | DeepSeek-R1+Graph | normal | 96.82 | 61.02 | 74.86 |
> | DeepSeek-R1+Graph | active | 5.27 | 55.31 | 9.62 |
> | DeepSeek-R1+Graph | recovered | 3.27 | 56.80 | 6.18 |

---

> ### Author Response · Authors · 2026-06-13
>
> **W2: Recommendation is trained and evaluated on observed user–food interactions, which may reward recommending foods users already ate rather than the healthier, clinically appropriate intervention foods the paper targets. The setup needs justification.**
>
> **R2:** The evaluation is **deliberately two-tiered, and it does not simply reproduce past consumption.**
> - Recall@20 and NDCG@20 measure **preference alignment** against held-out interactions, a precondition for adherence rather than our measure of clinical quality. H-Score@20, PA@20, and AvgTags@20 measure clinical **appropriateness and feasibility**. They are computed against condition-derived nutrition tags and price tags, independent of what the user consumed (Sec 4.2), so they do not reward already-eaten foods.
> - To test the concern directly, we add two metrics on the recommendation outputs. **Novel@20** is the fraction of top-20 items the user has never consumed anywhere in the data (train/val/test). **Novel-and-Appropriate@20** is the fraction that are both novel and satisfy at least one condition-derived tag. Replaying past consumption scores zero on both. Instead, Novel@20 is 0.91 for MOPI-HFRS and 0.99 for GAT, and Novel-and-Appropriate@20 is 0.37 and 0.29. Clinically appropriate recommendations are therefore **mostly novel foods rather than re-recommended consumption**. This rules out memorization as the source of constraint satisfaction.
> - The remaining gap between novelty and appropriateness reflects the **benchmark's central finding**: optimizing preference alignment does not by itself yield constraint satisfaction. CF methods raise Recall/NDCG with only marginal H-Score/PA gains, whereas constraint-aware models improve H-Score/PA without sacrificing ranking (Fig 4, Sec 4.2). Preference alignment is not clinical appropriateness, and **surfacing this decoupling is the purpose of GLEN-Bench**.
> - We use observed consumption because NHANES has **no ground-truth label** for the ideal intervention food. Separating preference proxies from consumption-independent constraint metrics is more faithful than synthesizing labels. Revisions: we reframe the task as multi-objective (preference-aligned, clinically appropriate, affordable), de-emphasize Recall/NDCG as intervention-quality measures, and note expert-defined targets as future work (Appendix G).
>
> **W3: The split is described as user-based, which may cause a cold-start issue for NGCF, LightGCN, SimGCL, SGL, and LightGCL. The paper should clarify how these methods handle unseen validation and test users, otherwise the comparison may be unfair.**
>
> **R3:** We apologize for the imprecise wording in Appendix B.2, which we have corrected. The concern does not arise in the actual implementation.
> - Recommendation does **not use a user-level split**. That sentence was mistakenly carried over from the detection-task protocol.
> - It uses a **standard transductive, interaction-level split**. The user–food edges are partitioned 60/20/20, so **every user and food node appears in the training graph and learns its ID embedding**. There is no cold-start, and all five methods are evaluated on users for which they have learned representations.
> - Inference is **leakage-free**. Message passing uses only training edges at validation, and training and validation edges at test.

---

> > ### Comment · Reviewer_NuFv · 2026-06-24
> >
> > Thanks for the thorough response! My concerns are all addressed.